# Epidermal progenitors suppress GRHL3-mediated differentiation through intronic polyadenylation promoted by CPSF-HNRNPA3 collaboration

Xin Chen[1,4], Sarah M. Lloyd [1,4], Junghun Kweon[1], Giovanni M. Gamalong [1] & Xiaomin Bao [1,2,3✉]

In self-renewing somatic tissue such as skin epidermis, terminal differentiation genes must be suppressed in progenitors to sustain regenerative capacity. Here we show that hundreds of intronic polyadenylation (IpA) sites are differentially used during keratinocyte differentiation, which is accompanied by downregulation of the Cleavage and Polyadenylation Specificity Factor (CPSF) complex. Sustained CPSF expression in undifferentiated keratinocytes requires the contribution from the transcription factor MYC. In keratinocytes cultured in undifferentiation condition, CSPF knockdown induces premature differentiation and partially affects dynamically used IpA sites. These sites include an IpA site located in the first intron of the differentiation activator GRHL3. CRISPR knockout of GRHL3 IpA increased full-length GRHL3 mRNA expression. Using a targeted genetic screen, we identify that HNRNPA3 interacts with CPSF and enhances GRHL3 IpA. Our data suggest a model where the interaction between CPSF and RNA-binding proteins, such as HNRNPA3, promotes site-specific IpA and suppresses premature differentiation in progenitors.

---

[1] Department of Molecular Biosciences; Weinberg College of Arts and Sciences, Northwestern University, Evanston, IL 60208, USA. [2] Department of Dermatology; Feinberg School of Medicine, Northwestern University, Chicago, IL 60611, USA. [3] Robert H. Lurie Comprehensive Cancer Center, Northwestern University, Chicago, IL 60611, USA. [4] These authors contributed equally: Xin Chen, Sarah M. Lloyd. ✉email: xiaomin.bao@northwestern.edu

Self-renewing somatic tissue, such as epithelium, undergoes continuous turnover to compensate for wear and tear. In this dynamic regeneration process, terminal differentiation is essential for fulfilling the specialized tissue function; However, terminal differentiation genes must be suppressed in tissue progenitors to sustain their regenerative capacity[1,2]. The molecular mechanisms underlying this spatiotemporal regulation of differentiation genes, in somatic tissue homeostasis, still remains incompletely understood.

Human skin epidermis, a stratified epithelium, is a highly accessible research platform for exploring gene regulatory mechanisms governing somatic tissue differentiation. Primary epidermal cells (keratinocytes) can be isolated and expanded in culture, in both undifferentiation and differentiation conditions, facilitating the integration of genomic, proteomic, and genetic tools into this platform[1–5]. Decades of research identified that epidermal differentiation involves the upregulation of multiple transcription factors, which bind to their chromatin targets and further activate barrier function[5,6]. One of these transcription activators is GRHL3, which is a selective late-differentiation activator that promotes protein crosslinking and cornified envelope formation[7–9]. GRHL3 is repressed in epidermal progenitors. Several distinct regulatory mechanisms, utilized by the progenitors to repress GRHL3, have been identified recently. These include PRMT1 binding at its promoter to influence its transcription, as well as EXOSC9 degrading mRNA post-transcriptionally[2,10]. In addition to the roles of transcription activators, epigenetic and post-transcriptional regulators as well as non-coding RNAs are also involved in regulating epidermal differentiation[1,6,11–18]. These findings highlight the coexistence of multiple gene regulatory mechanisms, at distinct steps of gene expression, to fine-tune the overall abundance of gene products.

About ~30% of the human genome is composed of introns[19]. Despite often being viewed as "junk", introns' influence on gene expression is being increasingly appreciated. Notably, polyadenylation sites have been identified in introns in addition to the well-established 3′UTR regions[20]. Usage of these intronic polyadenylation (IpA) sites terminates transcription prematurely. This IpA mechanism, although still under-studied, has been explored in several developmental and physiological processes. In muscle regeneration after injury, an IpA event in platelet-derived growth factor (PDGFRα) generates a short-isoform decoy to suppress full-length gene function and attenuate muscle fibrosis[21]. During the late stage of spermatogenesis, sterol regulatory element binding transcription factor 2 (SREBF2) switches from full-length to a short isoform using IpA to control germ-cell specific gene expression[22]. Recent transcriptome-wide profiling further identified recurrent IpA sites that inactivate tumor suppressor genes in leukemia[23], as well as the diverse IpA events in the immune system in response to various cellular environments[24]. How distinct IpA sites are being used in specific biological processes still remains unclear.

Polyadenylation requires cleavage of nascent RNAs, a process that involves the participation of multiple protein complexes. The cleavage stimulation factor (CstF) as well as the cleavage factors CF I and CFII bind to upstream and downstream elements[25]. The polyadenylation process is catalyzed by the polyadenylate polymerase (PAP). Notably, a central player of this process is the Cleavage and Polyadenylation Specificity Factor (CPSF) complex[26]. CPSF binds to the AAUAAA consensus sequence, the most important cis-regulatory element in cleavage and polyadenylation, through its CPSF4 and WDR33 subunits. CPSF3 is the endonuclease subunit that directly cleaves nascent RNA. These components, as well as two other regulatory subunits CPSF2 and FIP1L1, are brought together by a large scaffolding subunit CPSF1[27–30]. Intriguingly, the expression levels of CPSF subunits vary among different cell types[31]. Elevated CPSF expression was also observed in somatic cell reprogramming as well as in cancer[32,33]. How differential CPSF levels influence gene expression in somatic tissue, and what functions upstream to control CPSF expression, are still open questions.

In this study, we show that hundreds of IpA sites are differentially used by the transcription machinery during keratinocyte differentiation, which is accompanied by the reduction of CPSF expression. The enriched expression of CPSF in the progenitor state is downstream to MYC, and is essential for suppressing premature differentiation. We find that CPSF expression level influences a subset of differential IpA sites during keratinocyte differentiation. These CPSF-dependent IpA sites include an IpA site located in the first intron of GRHL3. Our CRISPR KO experiments show that the usage of this GRHL3 IpA site suppresses full-length GRHL3 mRNA expression. Using a combination of protein complex purification coupled with a genetic screen, we further identified HNRNPA3 as a key CPSF-interacting RNA-binding protein that enhances GRHL3 IpA usage. Taken together, our findings support a working model where CPSF cooperates with distinct RNA-binding proteins to modulate IpA usage in a site-specific manner, shaping the transcriptome of undifferentiated keratinocytes and suppressing premature differentiation.

## Results

**Keratinocyte differentiation involves altered IpA usage and downregulation of CPSF.** We leveraged the 3′READS+ technique, which features high sensitivity and strand specificity[34], to map transcriptome-wide PolyA sites in both undifferentiated and differentiated human keratinocytes. For every PolyA site identified in each gene, "Fraction of PolyA site Usage (FPU)" was calculated as the counts at this site divided by the sum of the counts from all PolyA sites in this gene (Supplementary Fig. 1a). With this quantification method, FPU is internally normalized within each sequencing library, as it calculates the fractions within individual genes. This method circumvents the technical challenge of normalizing PolyA site usage across different libraries, especially when gene expression is different between the two conditions. In total, 14625 PolyA sites were identified to robustly associate with 7990 expressed genes in keratinocytes (Fig. 1a). In undifferentiated or differentiated keratinocytes, the FPU of these sites is highly correlated between the replicates (Fig. 1b, c). The FPU is more divergent comparing undifferentiated versus differentiated keratinocytes (Fig. 1d), suggesting genome-wide changes in PolyA site usage between these two conditions. Among all these PolyA sites, 2739 of these sites are located in intronic regions (Supplementary Fig. 1b), 17% of which overlaps with the IpA sites recently identified in the immune system[24] (Supplementary Fig. 1c, d), highlighting the specificity of IpA events occurring in distinct tissue types. Both lists of IpA sites partially overlap with the PolyA_DB3 database[35] (Supplementary Fig. 1e, f). To identify the differentially used IpA sites between undifferentiated and differentiated keratinocytes, we integrated 3′READS+ data analysis with strand-specific RNA-seq data validation. In total, 428 differentially used IpA sites were identified (fold change > = 2 from 3′READS+, fold change ≥1.5 from RNA-seq, Fig. 1e, Supplementary Data 1). These IpA sites are highly enriched with the AAUAAA motif (E value: $1.9 \times 10^{-165}$). Notably, these differentially used IpA sites include more downregulated (60%, 256/428) than upregulated sites (40%, 172/428). Examples of these differentially used IpA sites include a downregulated IpA site in ESPN (cytoskeleton regulator), and an upregulated IpA site in the IQCK (EF-hand protein binding; Fig. 1f, g). Thus, dynamic IpA usage occurs during the cell-fate switch from the progenitor state to terminal differentiation.

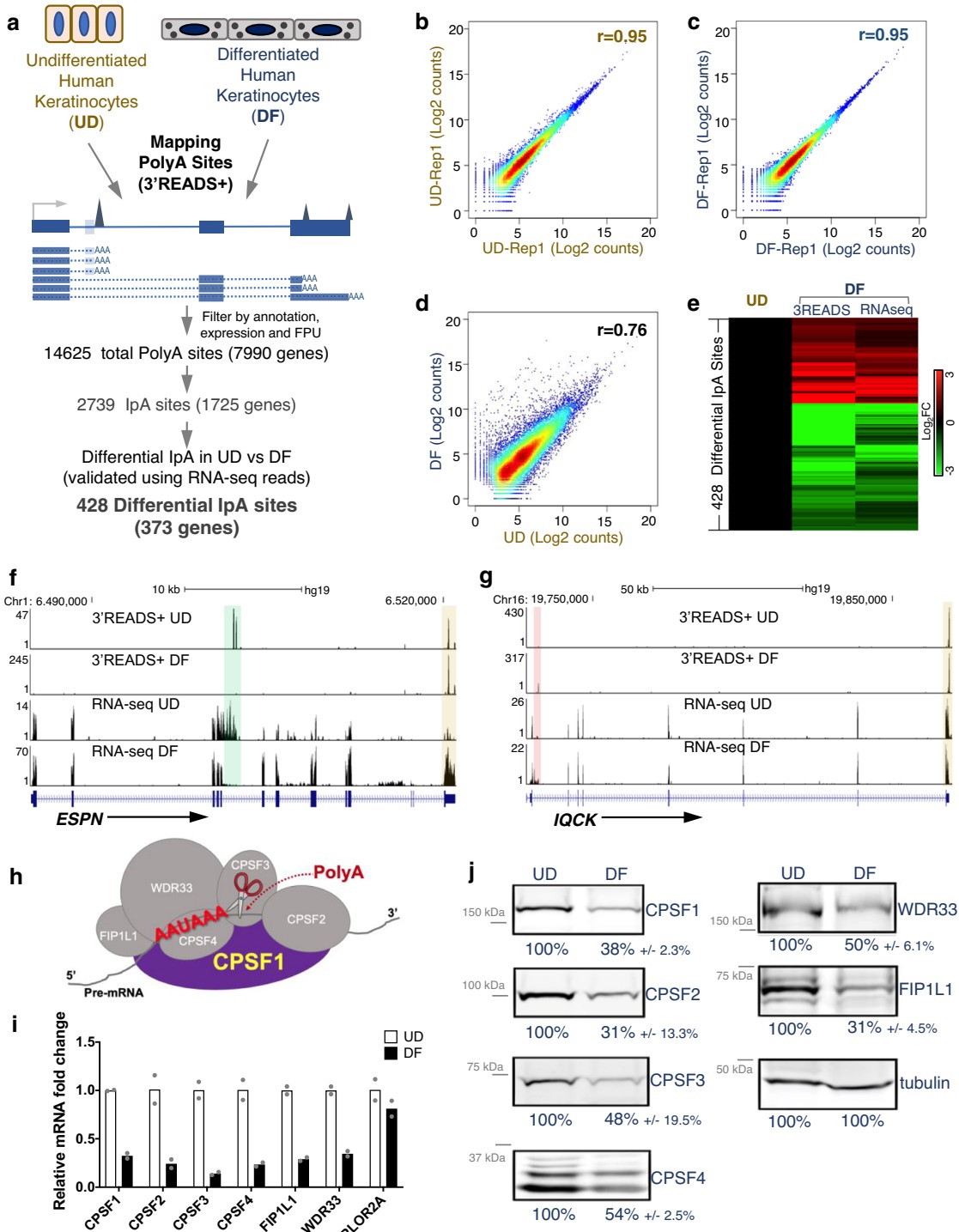

**Fig. 1 Keratinocyte differentiation involves IpA alterations and CPSF downregulation. a** Illustration showing the workflow of analyzing IpA sites in undifferentiated (UD) and differentiated (DF) keratinocytes. **b–d** Scatter plots showing the correlation of the 3′READS+ data between replicates, and between undifferentiated and differentiated conditions. **e** Heat map showing fold change of the 428 differentially used IpA sites during keratinocyte differentiation, in both 3′READS+ data and RNA-seq data. **f, g** Genome browser tracks showing the differential IpA usage in both EPSN and IQCK during keratinocyte differentiation, with both 3′READS+ and RNA-seq data. (Beige highlight: 3′UTR; Green highlight: UD-enriched IpA; Red highlight: DF-enriched IpA). **h** Illustration showing the composition and function of the CPSF complex. **h** Heat map comparing the relative mRNA levels of CPSF subunits and POLR2A between undifferentiated and differentiated keratinocytes. **i** qRT-PCR comparing the mRNA expression of CPSF genes between undifferentiated versus differentiated keratinocytes. Dots represent data points in technical replicates. **j** Immunoblots comparing the expression of CPSF subunits at the protein level between undifferentiated and differentiated human keratinocytes. β-tubulin was used as loading control. Average fold change and SD quantified from 3 replicates are indicated blow each panel. Source data are provided as a Source Data file.

In addition to these IpA sites, 11,506 sites were identified in 3′ UTR from our 3′READS+ analysis. 4677 of these sites correspond to genes with single PolyA sites in the 3′UTR, while 6829 sites were associated with 2727 genes that feature 2 or more PolyA sites in the 3′UTRs. To determine if 3′UTR shortening or lengthening could be a feature of keratinocyte differentiation, we calculated the fraction of distal PolyA site usage within 3′UTR of these 2727 genes, by taking the total counts from the most distal site divided by the sum of counts from the 3′UTR of that gene. This identified 211 genes with increased distal PolyA usage and 457 genes with decreased distal usage (fold change > 1.5, Supplementary Fig. 1g–i). These data indicate that both lengthening and shortening of 3′UTR occur during keratinocyte differentiation, although shortening occurs in more genes. This trend is similar to the findings in spermatogenesis where shortening of the 3′UTR was observed[36].

The polyadenylation process involves the participation of multiple complexes, including CPSF, CstF, CF I, CF II and PAP. The expression of PAP remains relatively constant between undifferentiated and differentiated keratinocytes, based on our RNA-seq data. Interestingly, the subunits encoding other complexes in this process are more dynamic (Supplementary Fig. 1j), including the downregulation of CPSF, the central player of cleavage and polyadenylation. Using both qRT-PCR as well as western blotting, the downregulation of core CPSF complex subunits in differentiation was confirmed at both mRNA and protein levels (Fig. 1h–j). Among the CPSF antibodies we used for Western blotting, the CPSF2 antibody worked in immunostaining of human skin sections, which exhibited stronger signals in the basal progenitor layer of the epidermis and reduced signal in differentiated layers. In comparison, the immunostaining of RNA polymerase II (Pol II) showed strong signals throughout both undifferentiated and differentiated layers of the epidermal tissue (Supplementary Fig. 1k). These data suggest that CPSF downregulation occurs during keratinocyte differentiation.

**Suppression of CPSF in undifferentiated keratinocytes induces differentiation.** The differential expression of CPSF raised the question of whether altered of CPSF level impacts keratinocyte function. Leveraging "ON TARGETplus" siRNA that simultaneously targets 4 different regions of CPSF1, we performed nucleofection in keratinocytes cultured in undifferentiated condition. A non-targeting control pool of four siRNAs was nucleofected in parallel as a negative control. The efficacy of CPSF1 RNAi was confirmed at both mRNA and protein levels (Fig. 2a and Supplementary Fig. 2a). In addition to CPSF1, the protein levels of other CPSF complex subunits were reduced. This was likely caused by protein degradation with loss of the CPSF1 scaffold, as the mRNA levels of these CPSF subunits were minimally affected (Supplementary Fig. 2b).

The regenerative capacity of keratinocytes with CPSF1 knockdown was assessed using a progenitor competition assay. In brief, epidermal tissue was regenerated using 50% keratinocytes expressing H2B-GFP and 50% of keratinocytes expressing H2B-mCherry. The H2B-GFP keratinocytes were treated with control siRNAs as an internal control; the H2B-mCherry keratinocytes were treated with CPSF1 siRNA or control siRNA. The regenerated epidermal was sectioned, and the ratio of mCherry-labelled nuclei versus GFP-labelled nuclei in the tissue sections was quantified using ImageJ[37]. CPSF1 knockdown strongly reduced the representation of mCherry-labelled cells as compared with control in the regenerated epidermal tissue (Fig. 2b, c). This reduction of mCherry-labelled cells was even more drastic in the bottom half of the epidermal tissue, as compared to the top more differentiated half (Supplementary Fig. 2c, d). These data suggest that CPSF1

knockdown impaired the regenerative capacity of keratinocytes. Consistent with the progenitor competition assay, epidermal tissue regenerated entirely using keratinocytes treated with CPSF1 RNAi was hypoplastic. The lipid stain Nile Red showed a significant increase in epidermal tissue with CPSF RNAi, suggesting enhanced barrier function to prevent water loss. No statistically significant difference was detected for apoptosis using the TUNEL assay (Supplementary Fig. 2e–h). In addition, CPSF RNAi also impaired keratinocyte migration in scratch wound healing assay (Supplementary Fig. 2i, j). Thus CPSF knockdown impaired multifaceted functions of keratinocytes critical for physiological processes such as tissue homeostasis and wound healing.

In addition to RNAi, we used CRISPRi[38] as an orthogonal method to suppress CPSF1 expression. This strategy involved the expression of the enhanced CRISPR repressor, which includes both KRAB and MeCP2 fused with the nuclease-inactive dCas9 for improved repression[39]. Three independent sgRNAs near the transcription start site (TSS) of CPSF1 were designed and expressed individually with dCas9-KRAB-MeCP2 in undifferentiated keratinocytes. All three independent sgRNAs successfully downregulated CPSF1 expression at both the mRNA and protein levels (Fig. 2d, e and Supplementary Fig. 2k). Using clonogenicity assay, the colony-forming ability of keratinocytes was assessed in CPSF CRISPRi versus non-targeting controls. The colony numbers were strongly reduced with CPSF1 loss (Fig. 2f, g), supporting that the intact function of CPSF is essential for epidermal progenitor self-renewal.

The validation of two knockdown approaches, RNAi and CRISPRi, allowed us to identify CPSF target genes that are altered in both. RNA-seq analysis identified 1113 genes from RNAi and 1584 genes from CRISPRi (fold change >=2, $p < 0.05$). The 739 genes shared in these two approaches are termed as "CPSF core targets" (Fig. 2h, i and Supplementary Data 2). The top enriched Gene Ontology (GO) terms of the upregulated core targets include "Keratinization" and "Keratinocyte differentiation", while the top GO terms of the downregulated genes include "DNA replication" and "Cell division" (Fig. 2j). These GO terms are very similar to the top GO terms associated with the genes altered during keratinocyte differentiation[6]. Using qRT-PCR, we validated several representative differentiation genes that were upregulated in CPSF suppression, including the differentiation activator GRHL3 as well as differentiation marker genes SPRR1B, S100A8, and S100A9 (Fig. 2k, l). These data identified an essential role of CPSF in suppressing premature differentiation in epidermal progenitor maintenance.

**Sustained CPSF expression in undifferentiated keratinocytes requires MYC.** As CPSF expression is downregulated in differentiated keratinocytes, we searched for mechanisms that could influence CPSF expression. We first asked if general impairment of proliferation and induction of differentiation in keratinocyte is sufficient to downregulate CPSF expression. Two independent strategies were tested. CPSF expression was first examined in keratinocytes with PRMT1 knockdown. PRMT1 was recently demonstrated as an essential regulator for sustaining proliferation and suppressing differentiation marker genes in undifferentiated keratinocytes[2]; however PRMT1 knockdown did not affect CPSF expression (Supplementary Fig. 3a). CPSF expression was also compared between early- and late-passage keratinocytes. Late-passage keratinocytes, after prolonged culture, expressed an increased level of differentiation markers such as p16 and S100A9, as well as reduced levels of cell cycle markers such as Ki67 and AURKB (Supplementary Fig. 3b). However, the expression of the CPSF subunits was not dramatically altered (Supplementary Fig. 3c). Thus, impaired proliferation and

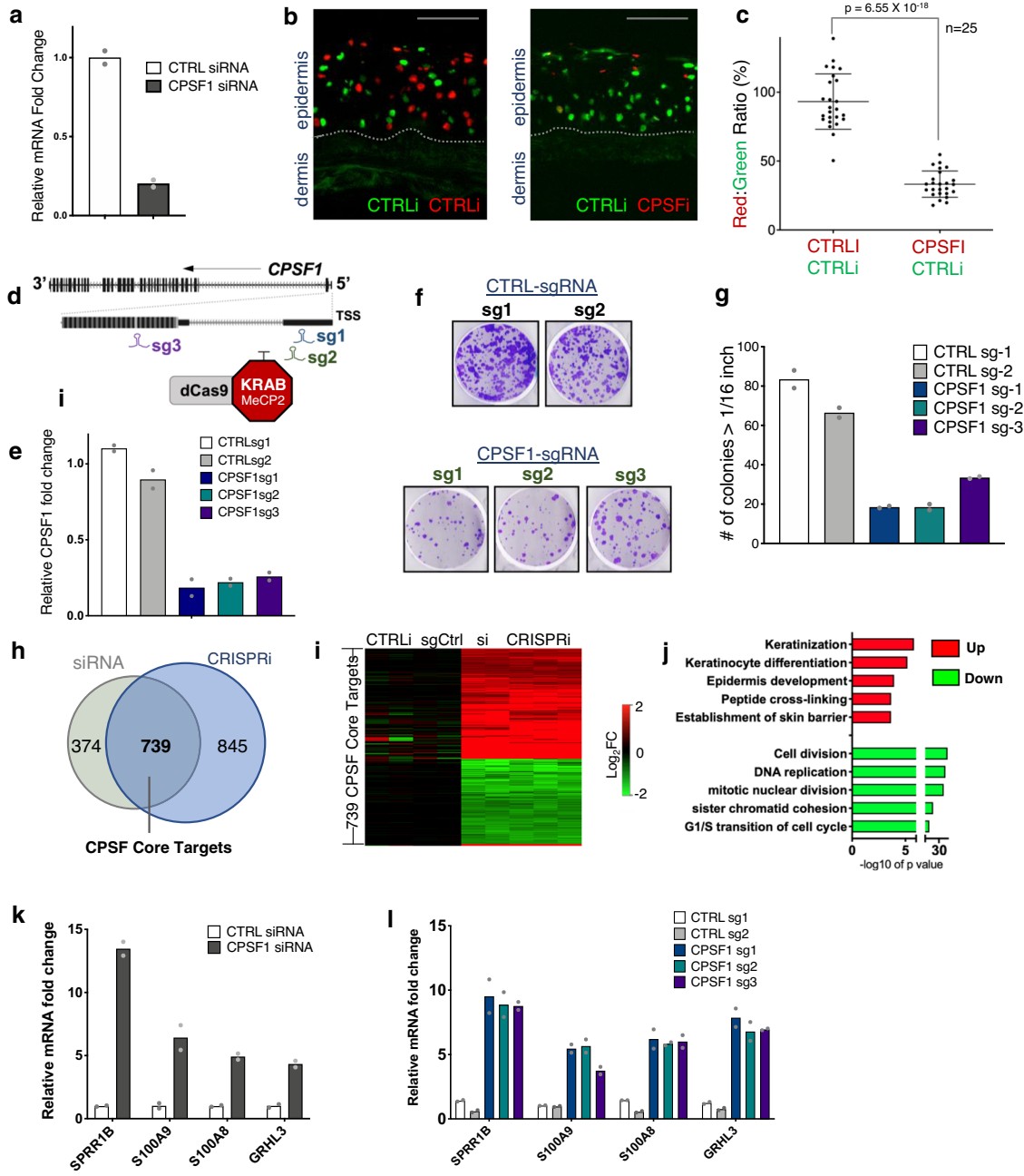

**Fig. 2 CPSF Downregulation in progenitors impairs self-renewal and induces differentiation. a** qRT-PCR quantification of CPSF1 knockdown efficiency. Dots represent data points in technical replicates. **b** Epidermal tissue regenerated by 50% CTRLi labelled by H2B-GFP (green), and 50% CTRLi or CPSFi labelled by H2B-mCherry (red). Scale bar: 100 µM. Representative images from 25 images per condition are shown. **c** Quantification of red:green ratio comparing tissue sections of CTRLi/CTRLi versus CTRLi/CPSFi ($n = 25$ images, $p = 6.55 \times 10^{-18}$, t-test, 2 tailed, error bars are presented as mean values ± SEM). **d** Diagram showing the design of CPSF1 CRISPRi. The locations of three independent sgRNAs (sg1, sg2, and sg3) are labelled relative to the transcription start site (TSS). **e** qRT-PCR quantification comparing the knockdown efficiency between CPSF CRISPRi versus control. Dots represent data points in technical replicates. **f**, **g** Clonogenic assay comparing keratinocytes with CPSF1 CRISPRi vs. non-targeting control. Colonies with diameter > 1/16 inch were quantified. Dots represent data points in technical replicates. **h** Venn diagram comparing the differentially expressed genes in CPSF siRNA vs. CPSF CRISPRi. These two data sets significantly overlap with each other (Fisher's exact test, 2-Tail, $p = 1.1 \times 10^{-308}$). **i** Heat map showing fold change of 739 differentially expressed genes (fold change > 2, $P < 0.05$, Wald test) shared between CPSF1 RNAi and CRISPRi. **j** Bar graph showing the top GO term associated with the genes significantly altered by CPSF CRISPRi (P values: modified Fisher's exact test). **k**, **l** qRT-PCR comparing mRNA levels of differentiation marker genes between CPSF1 knockdown vs controls. Dots represent data points in technical replicates. Source data are provided as a Source Data file.

induced differentiation are not sufficient to downregulate CPSF expression in primary human keratinocytes.

To identify specific regulators that could be essential for maintaining CPSF1 expression in undifferentiated keratinocytes,

we searched for transcription factors that can bind to the regulatory regions of the CPSF1 gene leveraging publicly available ChIP-seq data[40]. We found that the MYC ChIP-seq signal is enriched at the CPSF1 promoter, in both keratinocytes as well as

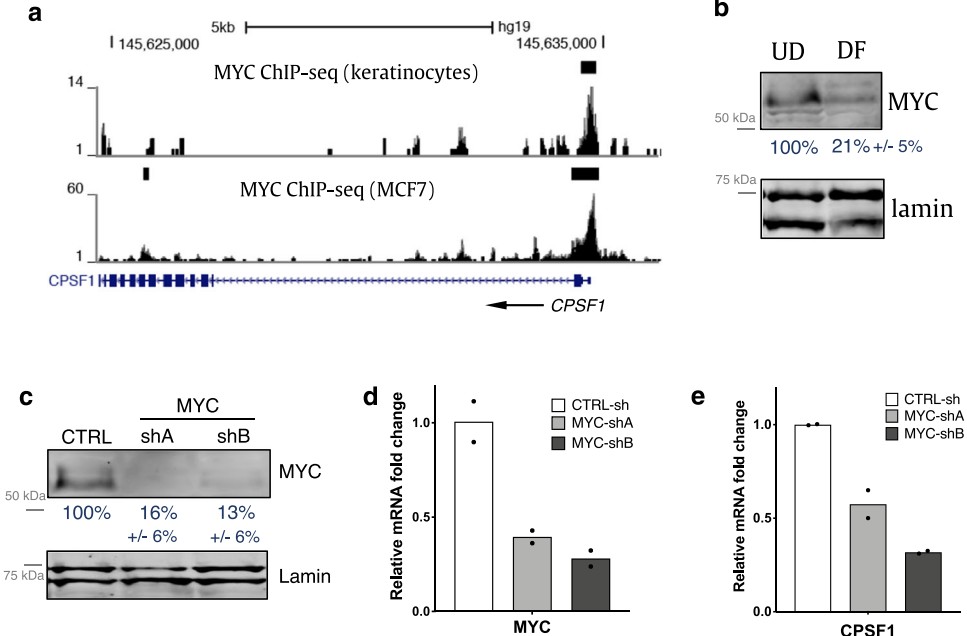

**Fig. 3 Sustained CPSF expression in undifferentiated keratinocytes requires MYC. a** Genome browser tracks showing the binding of MYC at CPSF1 promoter in keratinocytes as well as in MCF7 cells (GSE32883). **b** Western blots showing the downregulation of MYC during keratinocyte differentiation. Quantification of MYC protein level fold change is indicated below ($n = 3$). **c** Western blotting showing the knockdown efficiency of shRNAs targeting MYC. Quantification is indicated below ($n = 3$). **d, e** qRT-PCR quantification of MYC and CPSF1 comparing MYC knockdown versus control. Dots represent data points in technical replicates. Source data are provided as a Source Data file.

the breast cancer cell line MCF7 (Fig. 3a). MYC is downregulated during keratinocyte differentiation (Fig. 3b). To characterize the role of MYC in regulating CPSF expression, we first evaluated whether MYC could be essential for sustaining CPSF expression in undifferentiated keratinocytes. Two independent shRNAs targeting MYC were designed and validated using both qRT-PCR and Western blotting (Fig. 3c, d). Keratinocytes expressing these shRNAs showed downregulation of CPSF1, as compared to the non-targeting control shRNA (Fig. 3e). Since MYC is downregulated in differentiation, we also tested whether over-expression of MYC might be sufficient to increase CPSF expression in the differentiation condition. Between keratinocytes infected with the pCDH-MYC overexpression construct[41] versus the vector control, no drastic differences were observed in the mRNA level of CPSF1 (Supplementary Fig. 3d, e). These findings suggest that MYC is essential for sustaining CPSF expression in the progenitor state, yet it is not sufficient to drive high expression of CPSF in the differentiation state.

**CPSF downregulation alters IpA usage**. Since keratinocyte differentiation involves both altered usage of IpA sites as well as reduced CPSF levels, we tested whether CPSF knockdown could influence IpA usage using 3′READS+. We identified that 178 out of the 428 IpA sites differentially used in keratinocyte differentiation were altered in the same direction with CPSF knockdown (Fig. 4a, b). In particular, 74% of these IpA sites showed reduced usage in CPSF knockdown or in keratinocyte differentiation. Genes associated with these CPSF-dependent IpA sites include *ALOX15B, EPSN, CRBN*, as well as the differentiation activator *GRHL3* (Fig. 4c–f). Genes associated with CPSF dependent or independent IpA sites did not show drastic differences in gene expression (Supplementary Fig. 4a).

To validate the altered usage of these IpA sites, we developed a qRT-PCR strategy. In brief, two pairs of qPCR primers were used for each gene of interest, with the first pair (proximal) designed immediately before the IpA site, and the second pair (distal)

designed for exons between the IpA and the 3′ end of the gene. The relative usage of IpA can be quantified as proximal: distal ratio by using both pairs of primers in qRT-PCR. With this approach, we validated that the usage of these IpA sites in representative genes was strongly decreased in the context of keratinocyte differentiation, CPSF siRNA knockdown, CPSF CRISPRi, as well as MYC knockdown (Fig. 4g–j). Notably, the fold change of IpA usage for these IpA sites was less drastic in CPSF knockdown as compared to differentiation, indicating that the differentially used IpA sites in keratinocyte differentiation is partially influenced by CPSF downregulation. Usage of these IpA sites was minimally altered in migrating keratinocytes versus control (Supplementary Fig. 4b), suggesting that distinct biological processes occurring in the same cell type may involve different sets of IpA sites.

**CPSF suppresses GRHL3 expression by promoting IpA usage**. To explore whether altered IpA usage influenced by CPSF knockdown might be linked to gene expression, we intersected the CPSF core targets from RNA-seq with the 165 genes that are associated with the 178 CPSF-dependent IpA sites. This identified a total of 14 genes (Fig. 5a and Supplementary Data 3). The majority of these 14 genes show anticorrelation of fold change between RNA-seq and 3′READS+, in CPSF knockdown or in keratinocyte differentiation. Among them, the IpA site associated with GRHL3 stood out for a couple of reasons. First, this GRHL3 IpA site features the highest FPU in undifferentiated keratinocytes among all the IpA sites associated with these 14 genes (Fig. 5b), suggesting that the usage of this IpA site could play an important role in influencing the overall mRNA expression of its host gene. Second, GRHL3 is a transcriptional activator that can further modulate the expression of other epidermal differentiation marker genes. Using double knockdown with CPSF CRISPRi in combination with GRHL3 RNAi, we confirmed that GRHL3 RNAi suppressed the induction of a number of differentiation markers that were induced by CPSF CRISPRi alone, including

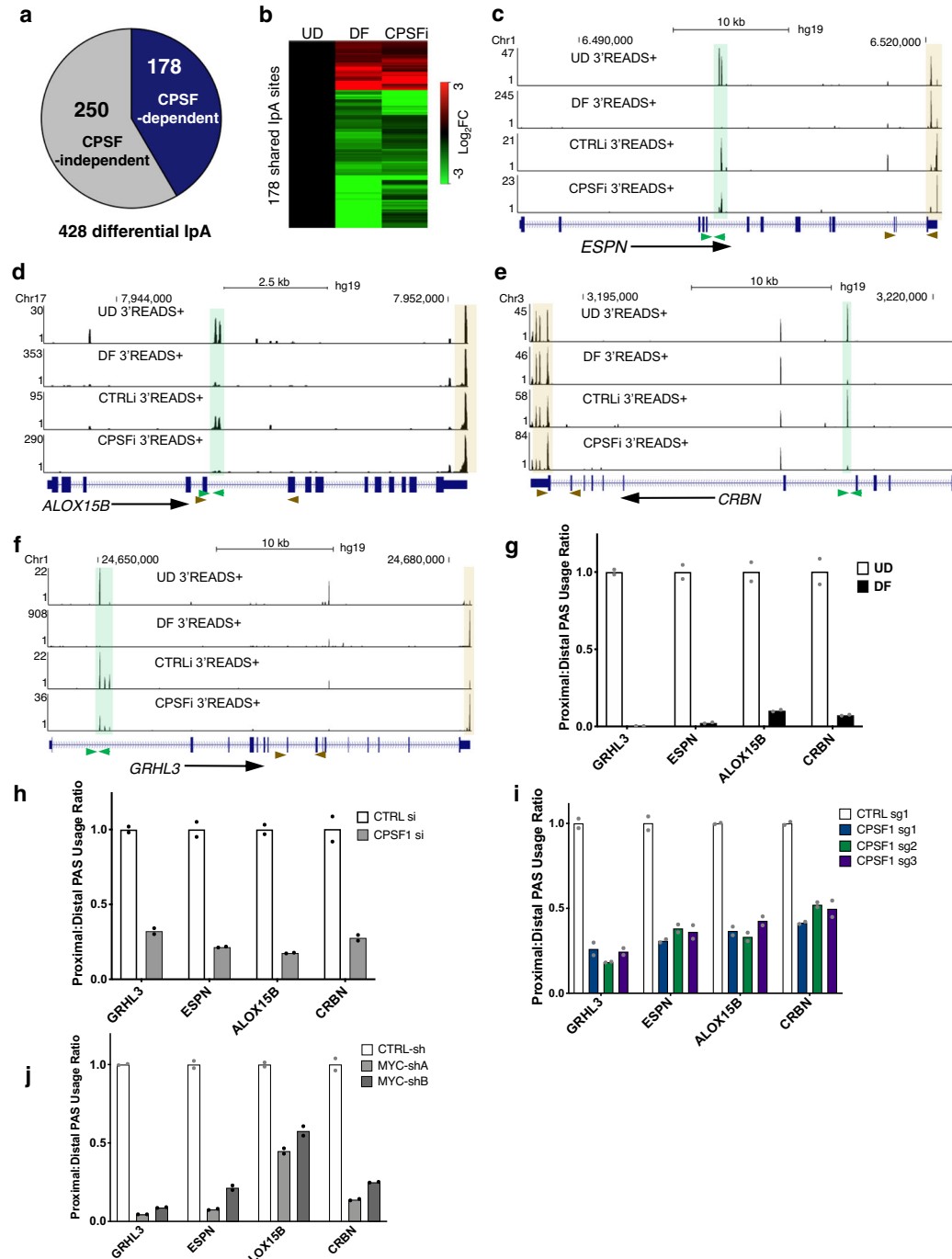

**Fig. 4 CPSF Downregulation Partially Accounts for the IpA Alterations in Differentiation. a** Pie chart showing 178 out of the 428 IpA sites altered in differentiation are influenced by CPSF1 knockdown. **b** Heat map showing the fold changes of these 178 IpA sites, comparing DF versus CPSFi. **c–f** Genome browser tracks comparing the differential IpA usage in DF and in CPSFi. Locations of the qPCR primers designed for IpA usage quantification are labelled below (brown arrowheads: proximal primers, green arrowheads: distal primers, beige highlight: 3′UTR, green highlight: UD-enriched IpA). **g–j** qRT-PCR quantification of the relative usage of the IpAs is calculated using the Proximal:Distal usage ratio, comparing undifferentiated vs. differentiated keratinocytes, CSPF1 siRNA and CPSF CRISPRi vs. control conditions, and MYC knockdown vs. control. Dots represent data points in technical replicates. Source data are provided as a Source Data file.

both mid-epidermal-differentiation markers (SPRR1B, S100A9, SPRR1A, S100A8) and late-epidermal-differentiation markers (SBSN and CRCT1; Fig. 5c–e). In particular, SBSN and CRCT1 were validated as direct targets of GRHL3, and the other four genes were also found to be downstream to GRHL3 in keratinocyte differentiation[42]. These data suggest that GRHL3 is a key downstream target mediating CPSF's role in suppressing

differentiation in epidermal progenitors, and that IpA could play a role in modulating GRHL3 expression.

This differentially used IpA site of GRHL3 is located in its first intron, about 4.3 kb downstream of the transcription start site. The usage of this IpA site is drastically downregulated in differentiated keratinocytes and in CPSF knockdown, while GRHL3 mRNA expression is strongly upregulated. To determine

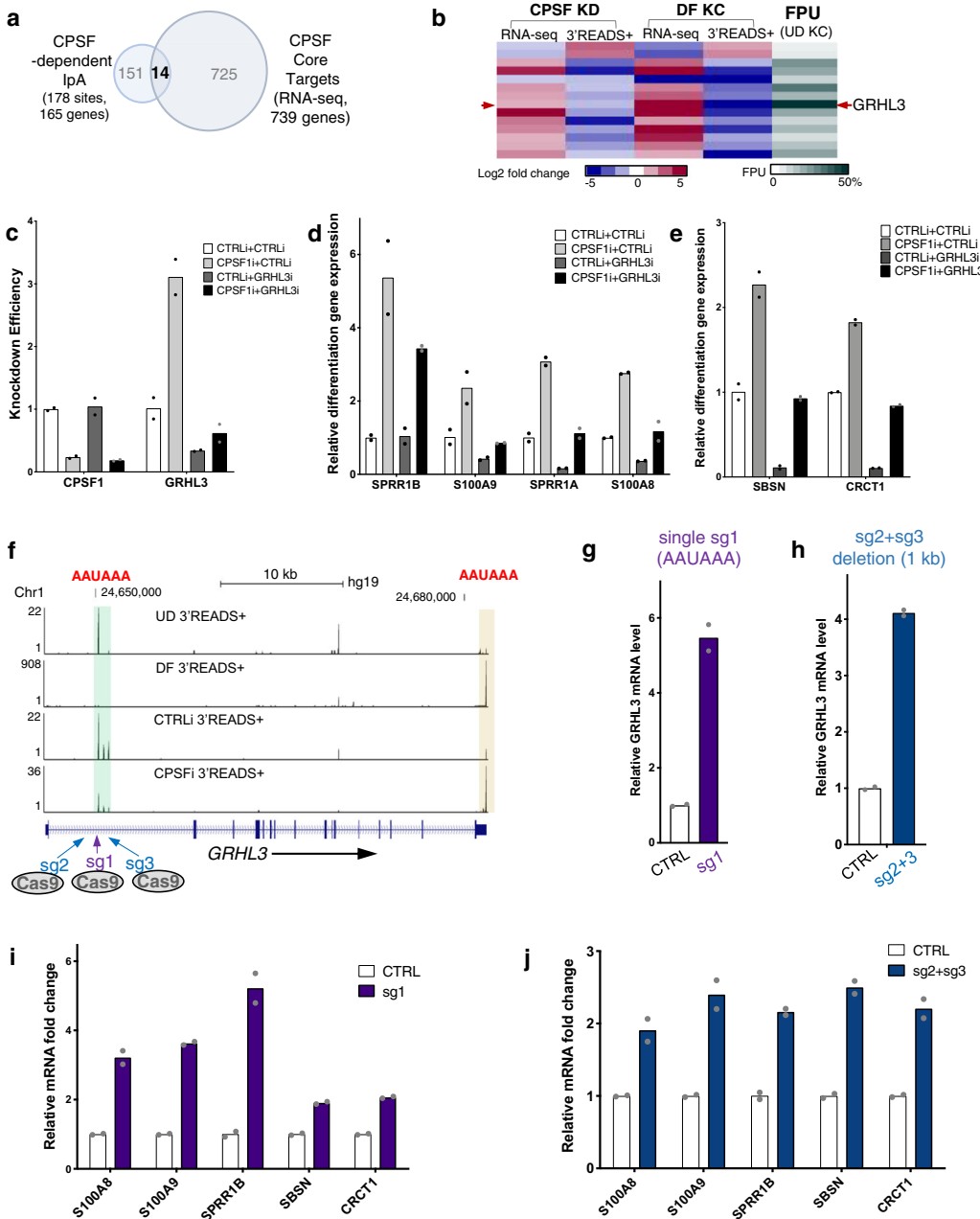

**Fig. 5 CPSF suppresses GRHL3 expression by promoting IpA usage. a** Venn diagram comparing genes associated with CPSF-dependent IpA sites and genes altered by CPSF knockdown. 14 genes are shared in these two lists of genes. **b** Heatmap showing the relative expression from RNA-seq, relative usage of IpA sites from 3′READS+, and the FPU of IpA in undifferentiated keratinocytes from 3′READS+. GRHL3 IpA has the FPU in undifferentiated keratinocytes. **c–e** qRT-PCR quantification of knockdown efficiency and differentiation marker gene expression in CPSF1-GRHL3 double knockdown. Downregulation of GRHL3 using siRNA in the context of CPSF1 CRISPRi largely restored the differentiation gene inhibition in keratinocytes. Dots represent data points in technical replicates. **f** Illustration showing the two CRISPR strategies to disrupt the GRHL3 IpA site. One strategy uses a single sgRNA (sg1) targeting the PAM site immediately adjacent to the AAUAAA consensus sequencing of the IpA. The second strategy uses two sgRNAs (sg2 and sg3) to delete 1-kb genomic sequence containing the IpA site. **g, h** qRT-PCR quantification of GRHL3 full-length mRNA level, comparing CRISPR KO vs control. Dots represent data points in technical replicates. **i, j** qRT-PCR quantification of differentiation marker gene expression comparing CRISPR KO vs control. Dots represent data points in technical replicates. Source data are provided as a Source Data file.

if usage of this IpA site could suppress the expression of full-length GRHL3, we took two different CRISPR approaches to knock out this IpA site (Fig. 5f). In the first approach, we leveraged a PAM sequence (AGG) directly adjacent to the AAUAAA (AATAAA of DNA sequence) CPSF-binding consensus sequence, and targeted this site with a single sgRNA (sg1) to create small indels. In a second approach, we co-expressed two sgRNAs (sg2 + sg3) that are designed to delete 1 Kb intronic

genomic sequence containing this IpA site. Both approaches achieved an average of 50% knockout efficiency in primary human keratinocytes, as estimated using PCR as well as the TIDE algorithm[43] (Supplementary Fig. 5a, b). As compared to non-targeting sgRNA controls, both approaches resulted in the upregulation of GRHL3 mRNA expression (Fig. 5g, h). These knockout cells also exhibited upregulation of differentiation marker genes that are downstream of GRHL3 (Fig. 5i, j). Thus

this genomic sequence of this GRHL3 IpA site, preferentially used in undifferentiated keratinocytes, plays a critical role in suppressing full-length GRHL3 gene expression and GRHL3-mediated differentiation.

Primary human keratinocytes only have a limited life span under the cell culture conditions without feeder cells. To further confirm the role of IpA in influencing GRHL3 expression, we expanded our scope to immortalized cell lines which allow isolation and expansion of single clones with CRISPR editing. Leveraging RNA-seq data of cell lines generated by the ENCODE project[44], we found that the GRHL3 IpA site is also used in HCT116 cells (Supplementary Fig. 5c). Similar to keratinocytes, these two CRISPR KO strategies upregulated of GRHL3 in bulk HCT116 cells (Supplementary Fig. 5d). We subsequently expanded and characterized 60 clones of HCT116 cells derived from the "sg2 + sg3" strategy, which allowed rapid PCR to screen deletion in one or both alleles. In total, 9 clones showed deletion in one of the two alleles (HET), and 1 clone showed deletion in both alleles (KO; Supplementary Fig. 5e). The HETs only displayed mild upregulation of GRHL3, while the KO showed drastic upregulation of GRHL3 at nearly 20 fold (Supplementary Fig. 5f, g). Sanger sequencing confirmed the expected deletion of ~1 kb containing the IpA site, created by the combination of sg2 and sg3. We also noticed minor differences (up to 50 bp at each end) among these individual clones, even between the two alleles within the single KO clone (Supplementary Fig. 5h). Taken together, these data demonstrate that the usage of this GRHL3 IpA site suppresses full-length GRHL3 mRNA expression.

**HNRNPA3 cooperates with CPSF to promote GRHL3 IpA**. CPSF downregulation alone could only partially explain the fold change and selectivity of the differentially used IpA sites during keratinocyte differentiation. In the example of GRHL3 IpA, the usage decreases more than 100-fold in differentiation, based on quantification by qRT-PCR. CPSF knockdown alone led to ~3–4-fold reduction of GRHL3 IpA, using the same quantification method. To identify other molecular mechanisms influencing IpA usage, synergistically with CPSF, a targeted screen was designed to identify potential cofactors enhancing GRHL3 IpA. As illustrated in Fig. 6a, this strategy involved double knockdown of CPSF in combination with a candidate cofactor, to determine if double knockdown reduces GRHL3 IpA usage more than CPSF knockdown alone.

Putative CPSF-interacting proteins were included as candidates in this genetic screen, based on our pilot mass spectrometry experiment comparing CPSF1 immunoprecipitation between undifferentiated and differentiated keratinocytes (Supplementary Fig. 6a, b). This experiment identified several proteins related to the function of RNA binding, associating with CPSF1 in the undifferentiated but not in the differentiated condition. In addition, we also looked into complexes that cooperate with CPSF in cleavage and polyadenylation, such as CSTF, CFI and CFII. We prioritized CSTF2 as a target in this screen, as CSTF2 was the most downregulated subunit of these complexes, according to our RNA-seq data of keratinocyte differentiation.

Two shRNAs targeting each candidate gene were validated with their knockdown efficiency (Fig. 6b), and were introduced to keratinocytes in combination with CPSF siRNA. HNRNPA3 showed the highest reduction of GRHL3 IpA usage in double knockdown versus single knockdown (Fig. 6c, d). Using co-immunoprecipitation, we confirmed that HNRNPA3 associated with CPSF1 only in the undifferentiated condition (Fig. 6e). HNRNPA3 is expressed in both undifferentiated and differentiated keratinocytes, although the protein level is slightly reduced in differentiation (Supplementary Fig. 6c). Thus, the reduced interaction between CPSF1 and HNRNPA3 in differentiation could be a result of the reduced expression of both proteins. In undifferentiated keratinocytes, the association between CPSF1 and HNRNPA3 was not disrupted by RNase (Fig. 6f), suggesting that HNRNPA3 and CPSF1 do not require RNA to bridge their interaction. The usage of other differential IpA sites, such as the sites associated with ALOX15B and CRBN (Supplementary Fig. 6d, e), were not drastically enhanced by HNRNPA3 knockdown in the context of CPSF RNAi. These data suggest that the differentially used IpA sites in keratinocyte differentiation are regulated by diverse mechanisms, and HNRNPA3 selectively influences a subset of IpA sites such as GRHL3 IpA.

**HNRNPA3 suppresses keratinocyte differentiation and influences GRHL3 splicing**. The cooperation between HNRNPA3 and CPSF1 in controlling GRHL3 IpA suggests that they could synergistically suppress differentiation in epidermal progenitors. RNA-seq data comparing HNRNPA3 knockdown versus control identified a total of 1490 differentially expressed genes (fold change >=2, $p < 0.05$, Supplementary Data 4 and Fig. 7a, b). The upregulated genes are highly enriched with GO terms such as "epidermal development" and "keratinocyte differentiation" (Fig. 7c). A total of 306 genes are affected by the downregulation of HNRNPA3 and CPSF1 (Fig. 7d). These overlapping genes include GRHL3 as well as GRHL3 target genes, such as SPRR1b and SBSN. Double knockdown of GRHL3 and CPSF drastically elevated the expression of these differentiation genes, as compared to single knockdowns (Fig. 7e–g). Thus, HNRNPA3 cooperates with CPSF to suppress a subset of differentiation genes, although HNRNPA3 also has CPSF-independent roles in gene regulation.

We next explored the nature of this HNRNPA3-CPSF collaboration in controlling GRHL3 IpA. HNRNPA3 is part of the hnRNP A/B family. The proteins in this family are characterized by two tandem RNA recognition motifs (RRMs) in the N-terminal region[45,46]. Systematic analysis of their *Drosophila* homologs indicates that these hnRNP A/B proteins control overlapping but diverse targets in pre-mRNA processing[47]. The best-characterized human protein in this family is HNRNPA1, which binds to the consensus sequence UAGGGA/U and directly antagonizes splicing[48,49]. HNRNPA3 shares 94% similarity in the tandem RRMs with HNRNPA1, but differs in the C-terminal region. Although the function of HNRNPA3 is currently under-characterized, the high similarity of RRMs suggests that HNRNPA3 and HNRNPA1 could bind to a very similar RNA consensus sequence to influence pre-mRNA processing.

RNA-seq data in undifferentiated keratinocytes, generated by us as well as by previous studies[50], identified a potential "hidden exon" with enriched RNA-seq reads immediately before the GRHL3 IpA site (Fig. 7h), suggesting that GRHL3 IpA may involve the inclusion of this "hidden exon" through alternative splicing. To quantify potential alternative splicing events including or skipping this "hidden exon", qPCR primers were designed to amplify the junction between exon 1-"hidden exon" versus exon1-exon2 (Fig. 7i). HNRNPA3 knockdown drastically increased the ratio of skipping versus inclusion of the "hidden exon" (Fig. 7j), suggesting that HNRNPA3 antagonizes the splicing between exon1-exon2 and promotes the connection between exon1-"hidden exon". To further clarify the role of HNRNPA3 in splicing, additional primer pairs were designed to quantify the ratio between exon1-intron1 junction versus exon1. This ratio was strongly reduced in HNRNPA3 knockdown, but not in CPSF1 knockdown (Fig. 7k, l), suggesting that HNRNPA3 stabilizes the exon1-intron1 junction and suppresses splicing of intron 1.

Taken together, these findings suggest a working model where CPSF and RNA-binding proteins cooperatively influence gene

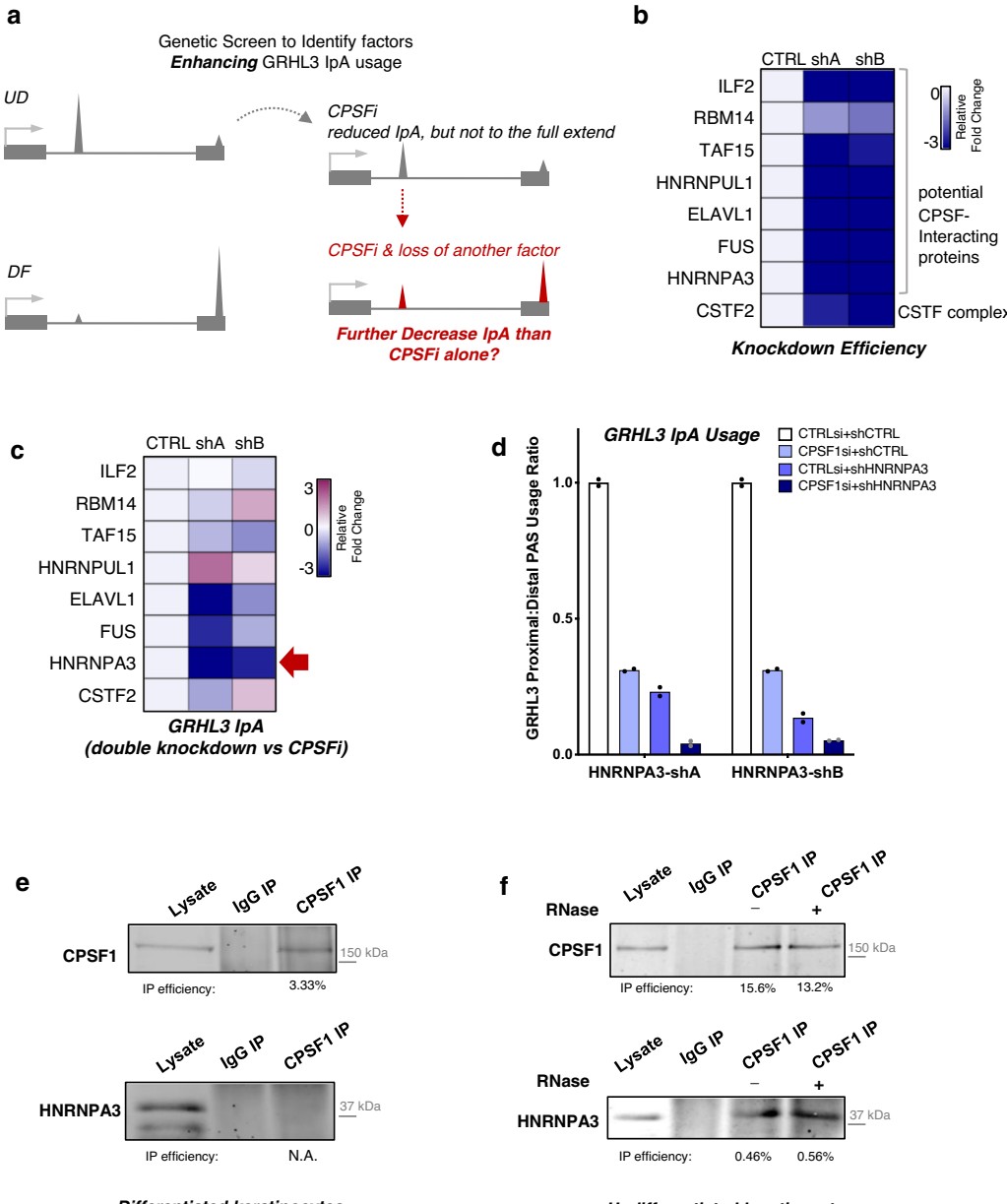

**Fig. 6 HNRNPA3 cooperates with CPSF to promote GRHL3 IpA. a** Illustration showing the genetic screen for identifying factors that enhance GRHL3 IpA. **b** Heat map showing the knockdown efficiency of the two shRNAs targeting specific candidates that include potential CPSF1-interacting proteins as well as CSTF2. **c** Heat maps showing the relative fold change of GRHL3 IpA usage comparing double knockdown vs CPSF single knockdown. **d** qRT-PCR showing the fold change of GRHL3 IpA usage, comparing HNRNPA3-CPSF double knockdown, single knockdowns and control. Dots represent data points in technical replicates. **e**, **f** Co-immunoprecipitation showing CPSF1 associates with HNRNPA3 in undifferentiated condition, but not in differentiated condition. This association is not disrupted by RNase treatment. Quantification of immunoprecipitation efficiency is indicated below. Source data are provided as a Source Data file.

expression through controlling the usage of specific IpA sites. In the context of epidermal tissue progenitors, MYC functions upstream of CPSF to sustain its high level of expression. The physical interaction between CPSF and HNRNPA3 synergistically promotes GRHL3 IpA, through suppressing the junction between the adjacent exons and promoting cleavage of nascent RNA, to suppress the premature expression of terminal differentiation genes in epidermal progenitor maintenance (Fig. 8).

## Discussion

It remains incompletely understood how genomic information is being selectively accessed, to fine-tune spatiotemporal gene regulation in development and in somatic tissue homeostasis. In this study, we found that human keratinocyte differentiation involves differential usage of intronic polyadenylation sites. In particular, we characterized an IpA site located within the first intron of GRHL3, a key transcriptional activator of epidermal differentiation. Usage of GRHL3 IpA in epidermal progenitors contributes to suppressing the expression from GRHL3 as well as the terminal differentiation genes downstream to GRHL3. Both CPSF and its interacting protein HNRNPA3 are essential for promoting the usage of this IpA site.

HNRNPA3 is one of the several RNA-binding proteins which were identified to associate with CPSF in undifferentiated, but not differentiated keratinocytes. In the genetic screen,

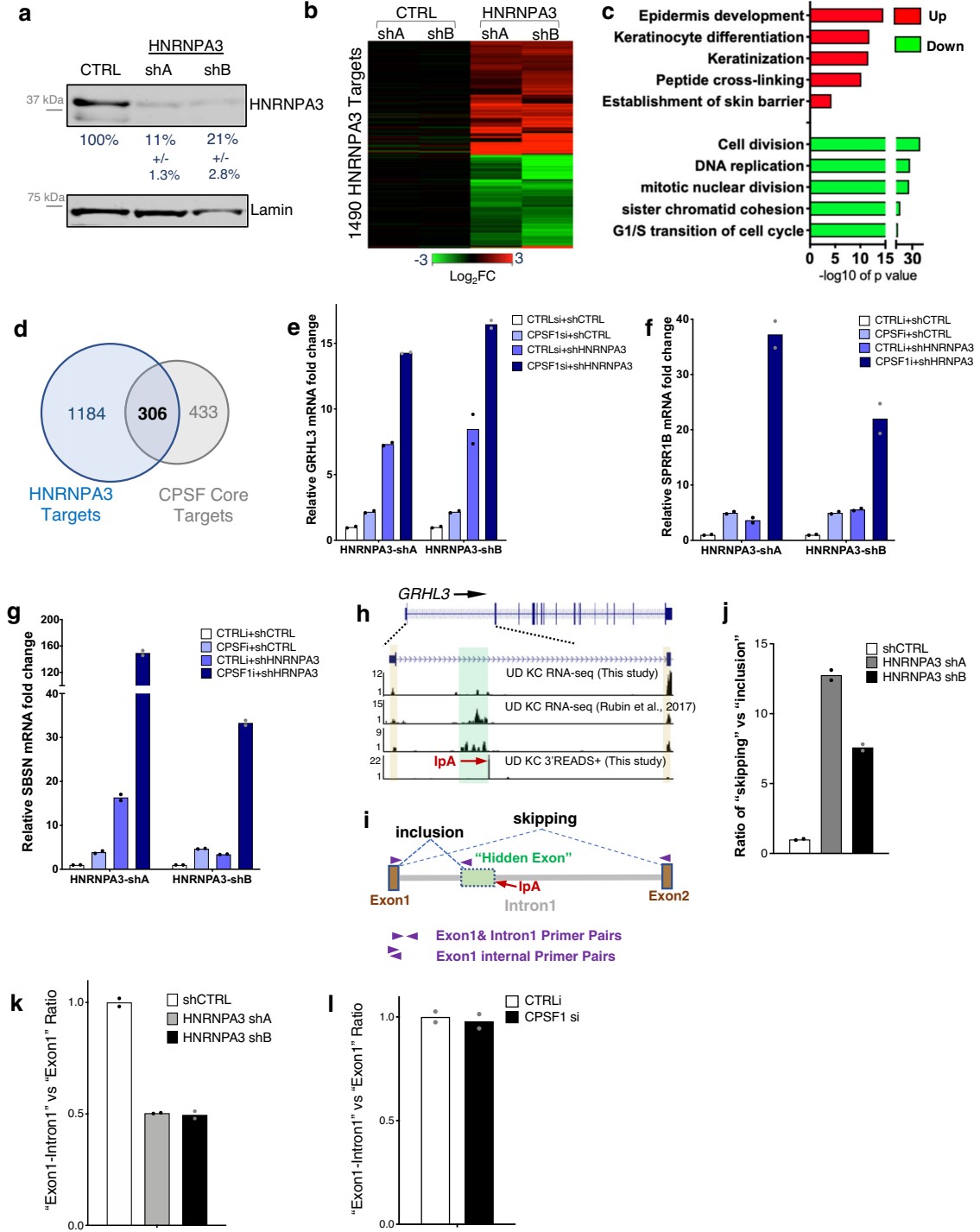

**Fig. 7 HNRNPA3 suppresses differentiation and influences GRHL3 splicing. a** Western blotting validating the knockdown efficiency of two shRNAs targeting HNRNPA3, at the protein level. Quantification is indicated below ($n = 2$). **b** Heat map showing the relative expression of 1490 genes that are differentially expressed between HNRNPA3 knockdown versus control. **c** Bar graph showing the top GO terms associated with these 1490 genes altered by HNRNPA3 knockdown (p value: modified Fisher's exact test). **d** Venn diagram comparing HNRNPA3 target genes versus CPSF core target genes. **e–g** qRT-PCR quantification of GRHL3, SPRR1B and SBSN in double knockdown and single knockdown of HNRNPA3 and CPSF1. Dots represent data points in technical replicates. **h** RNA-seq and 3'READS + tracks showing enriched RNA-seq reads before the *GRHL3* IpA site. These RNA-seq reads are likely to be enriched in a "hidden exon". **i** Illustration showing the qRT-PCR strategies to quantify "inclusion" and "skipping" of the hidden exon, as well as the ratio of pre-mRNA versus total RNA. Arrow heads indicate the location of the primers. **j** Quantification of the relative ratios of "inclusion" versus "skipping" of the "hidden exon" comparing HNRNPA3 knockdown versus control. HNRNPA3 knockdown promoted the splicing between exon1 and exon2 and skipped the "hidden exon". Dots represent data points in technical replicates. **k**, **l** Ratio of qRT-PCR amplification between Exon1-intron1 junction versus exon1, in HNRNPA3 knockdown or CPSF1 knockdown. Dots represent data points in technical replicates. Source data are provided as a Source Data file.

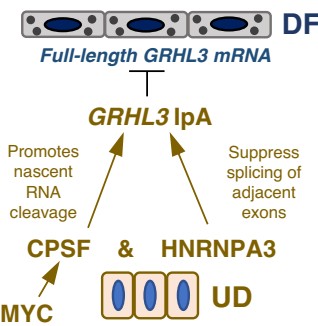

**Fig. 8 Working model.** Graphic illustration showing the working model of CPSF-HNRNPA3 cooperation in epidermal progenitor maintenance. MYC functions upstream to sustain CPSF expression in undifferentiated keratinocytes. In the first intron of GRHL3, CPSF and HNRNPA3 bind to nascent RNA to cooperatively suppress splicing and promote intronic polyadenylation, which suppresses full-length GRHL3 mRNA expression and inhibit premature differentiation. UD undifferentiated keratinocytes, DF differentiated keratinocytes.

HNRNPA3 stood out to have the strongest influence to enhance CPSF's ability to suppress GRHL3 IpA. Although the molecular function of HNRNPA3 is not fully understood, its high homology to HNRNPA1 suggests its role in splicing. In the case of HNRNPA1, multiple copies of this protein can bind and spread on nascent RNA to suppress splicing by altering RNA secondary structure and displacing splicing-promoting proteins such as the serine/arginine (SR)-rich-family proteins[51,52]. Our qPCR quantification identified that the HNRNPA3 knockdown promoted exon1-exon2 junction and destabilized exon1-intron1 junction. In addition, motif search using RBPmap[53] identified three HNRNPA1 motif sites ($p < 0.01$, Z-score > 2.5) within 2 kb upstream of the GRHL3 IpA site, suggesting that HNRNPA3 could bind to GRHL3 pre-mRNA and influence IpA usage through splicing. These data suggest that HNRNPA3 can involve at least two mechanisms to influence GRHL3 splicing and polyadenylation: HNRNPA3 can bind directly to pre-mRNA within the first intron to suppress the splicing between exon1 and exon2; HNRNPA3 can also bind to CPSF to stabilize CPSF binding to the AAUAAA motif, facilitating the full assembly of cleavage and polyadenylation machinery to promote IpA. Our "sg2 + sg3" CRISPR KO strategy, designed to remove the AAUAAA motif, did not affect these three putative HNRNPA3 binding sites. Future studies dissecting the contributions from HNRNPA3 in regulating this GRHL3 IpA, using additional CRISPR strategies to KO these three putative HNRNPA3 binding sites, can further elucidate the contribution from both HNRNPA3 and CPSF to regulating GRHL3 expression.

In addition to HNRNPA3, a couple of other RNA binding proteins such as FUS and ELAVL1, also enhanced GRHL3 IpA usage in our screen, although to a lesser extent as compared to HNRNPA3. FUS had been previously demonstrated to bind to nascent RNA near the alternative PolyA sites[54]. ELAVL1 is also implicated in alternative splicing. For example, ELAVL1 loss was known to promote exon 11 skipping of the translation initiation factor *Eif4enif1*[55]. ELAVL1's association with TRA2-beta was demonstrated to promote the inclusion of exon2[56]. Therefore the selectivity of GRHL3 IpA is likely to involve the cooperation from multiple RNA-binding proteins, although HNRNPA3 had the strongest effect from our target screen. As HNRNPA3 did not appear to enhance CPSF1's regulation of ALOX15B IpA and CRBN IpA, the usage of different IpA sites may involve diverse regulatory mechanisms.

Previous studies in different systems demonstrate that IpA can lead to truncated proteins[21,23,24]. In the case of GRHL3, this specific IpA is located within the first intron. If translated, this isoform would only retain 6 amino acids of the original GRHL3 protein. For the other IpA sites, their relative location varies among different genes. For example, the ALOX15B IpA is located in the 5th intron. The mRNA generated through ALOX15B IpA could be translated into a protein that misses ~80% of the lipoxygenase domain. A key technical barrier at present, for characterizing the roles of additional IpA sites, is the lack of high-quality antibodies that are specifically raised to target the N-terminal regions of these proteins.

Among the 2739 IpA sites that we identified in keratinocytes, 17% of these sites overlap with the IpA sites cataloged in the immune system[24]. However, when we compared the host genes associated with these IpA sites in these two systems, the overlap increased to 45%. For example, in keratinocytes the AGO3 gene is associated with 3 IpA sites, and in the immune cells AGO3 is associated with 4 IpA sites. Only 2 of these IpA sites overlap. Thus the same gene can associate with shared and distinct IpA sites in different cell types.

In summary, our work provides genome-wide profiling of polyadenylation in human keratinocyte differentiation, and sheds light into the regulatory mechanisms underlying the usage of specific IpA sites. This work also reveals the essential roles of CPSF and HNRNPA3 in regulating keratinocyte differentiation, highlighting the significance of pre-mRNA processing in influencing somatic tissue homeostasis.

## Methods

**Cell culture.** Primary human keratinocytes were isolated from the surgically discarded fresh foreskin (obtained from Northwestern Skin Biology & Diseases Resource-Based Center). Tissue was collected under a protocol approved by the Northwestern University Institutional Review Board (IRB # STU00009443). Patient consent for neonatal foreskin tissue was not required as this tissue is de-identified and considered discarded material per IRB policy. Keratinocytes from at least three de-identified donors were mixed and cultured in 50% complete Keratinocyte-SFM (Life Technologies #17005-142) and 50% Medium 154 (Life Technologies #M-154-500). Keratinocyte differentiation was induced by adding 1.2 mM $CaCl_2$ in full confluency for four days. HEK293T and phoenix cells were cultured in DMEM (Gibco) containing 10% fetal bovine serum (HyClone). HCT116 cells were cultured in McCoy's 5A (Modified) Medium (Gibco) containing 10% fetal bovine serum.

**Plasmid construction.** For lentiviral CRISPRi, the pLEX_Cas9 plasmid (Addgene #117987) was modified by replacing its Cas9 sequence with KRAB-dCas9 from pHR-SFFV-KRAB-dCas9-P2A-mCherry (Addgene #60954) to make pLEX-KRAB-dCas9-BSD. Then its CMV enhancer and promoter were replaced by the UCOE (Ubiquitous Chromatin Opening Element)-SFFV promoter from pMH0001 (Addgene #85969) to generate pLEX-UCOE-SFFV-KRAB-dCas9-BSD.

For retroviral CRISPRi, dCas-KRAB-MeCP2-BSD was PCR amplified from the pB-CAGGS-dCas9-KRAB-MeCP2 plasmid (Addgene #110824), and then cloned into the retroviral vector LZRS linearized by BamHI and NotI.

For retroviral CRISPR, Cas9-BSD was PCR amplified from the pLEX_Cas9 plasmid (Addgene #117987), and then cloned into the retroviral vector LZRS linearized by BamHI and NotI.

For sgRNA cloning, we linearized the pLentiGuide plasmid (Addgene #117986) by BsmBI, and ligated sgRNA sequence into it to make pLentiGuide-sg-mCherry.

For GRHL3 and ESPN tandem sgRNA cloning, a double-stranded DNA block containing the sgRNA2-tRNA-sgRNA3 was synthesized from GENEWIZ, Inc., and then cloned into the pLentiGuide plasmid (Addgene #117986) linearized by BsmBI.

For shRNA cloning, we linearized pLKO.1 puro plasmid (Addgene #8453) by AgeI and EcoRI, and ligated annealed shRNA oligos into it to make pLKO.1-sh-puro. The oligo sequences for generating shRNA constructs are included in Supplementary Data 5.

**Gene transfer.** Transfection of HEK293T and phoenix cells was performed using Turbofect (Thermo Fisher) following the manufacturer's instruction. High-titer virus was collected at 48 and 72 h post-transfection, added to wells of keratinocytes and centrifuged at 1250 rpm for 1 h at 32 °C. Keratinocytes were then selected using blasticidin (5 µg/mL) or puromycin (2 µg/mL) after infection for 48 h. HCT116 cells were virally infected and selected for blasticidin or puromycin resistance using the same methods as keratinocytes.

**siRNA knockdown.** ON-TARGETplus siRNA- SMARTpool targeting CPSF1 (L-020395-00) and GRHL3 (L-014017-02) were ordered from Dharmacon. 4D-

Nucleofector (Lonza) was used for nucleofection following the manufacturer's instruction.

**Genomic DNA knockout analysis.** In order to check CRISPR knockout efficiency of GRHL3 IpA in keratinocytes and HCT116 cells, total genomic DNA was extracted using Quick-DNA Miniprep Plus Kit (Zymo). For sg-1 disruption, the corresponding genomic region harboring the knockout site was amplified using Phusion High-Fidelity DNA Polymerase (Thermo Fisher), and gel extracted for Sanger sequencing (ACGT, INC.). Sequencing results for the mixed pool were analyzed using TIDE[43] webtool (https://tide.deskgen.com/). For sg2-sg3 deletion, the corresponding genomic region harboring the knockout site was amplified using GoTaq® DNA Polymerase (Promega), resolved on an agarose gel, and quantified using Image J software (NIH). For analysis of the two alleles in the single-clone KO HCT116 cells, the PCR product was cloned into the pLZRS vector and individual clones were analyzed using Sanger sequencing.

**Protein expression and tissue analysis.** For immunoblot analysis, 20–50 μg of cell lysate was loaded per lane for SDS-PAGE and transferred to PVDF membranes. The blots were scanned and quantified using the Li-COR Odyssey Clx imaging system (LI-COR). For immunofluorescence staining, tissue sections (7 μm thick) were fixed using either 50% acetone and 50% methanol, or 4% formaldehyde. Primary antibodies were incubated at 4 °C overnight and secondary antibodies were incubated at room temperature for 1 h. Images were captured by an EVOS FL Auto 2 fluorescent microscopy (Thermo Fisher) and processed by Image J software (NIH).

**Colony formation assay.** Mouse fibroblast 3T3 cells were treated with 15 μg/mL mitomycin C (Sigma) in serum-free DMEM for 2 h, then trypsinized and plated at ~8 × 10⁵ cells per well in a 6-well plate overnight. The media was changed to FAD media 1 h before seeding 1000 keratinocytes onto the feeder layer. The medium was changed every 2 days for eight days. Then the wells were washed with PBS to remove the 3T3 cells, and remaining keratinocytes were fixed in 1:1 acetone/methanol for 5 min. The plates were allowed to air dry for 5 min, and then colonies were stained with crystal violet.

**Wound healing assay.** In 24-well plates, 0.2 × 10⁶ CTRL or CPSF1 knockdown keratinocytes were seeded and allowed to grow to full confluence the next day. Keratinocytes were treated with 10 μg/mL mitomycin C for 2 h prior to wounding. Wounds were made using 100-μl filter pipette tip (Thermo Fisher) and the healing process was monitored under a microscope. For quantification, the surface area of the scratch at different time points was measured using the Image J software (NIH). For comparing IpA usage between migrating keratinocytes versus control, a grid of scratches were made to a confluent monolayer of keratinocytes as described previously[9]. RNA extraction and qPCR analysis were performed using scratched versus non-scratched plates.

**Organotypic human epidermis regeneration.** For organotypic epidermal cultures, keratinocytes were nucleofected with CPSF1 siRNA or non-targeting control siRNA, trypsinized 4 days later and counted. The dermis was prepared from donated cadaver samples from the New York Firefighters Skin Bank. Usage of human dermis from de-identified donors for organotypic epidermal regeneration has been approved by Northwestern IRB. In all, 1.0 × 10⁶ CTRL or CPSF1 knockdown keratinocytes were seeded onto the top of each piece of the devitalized dermis. The organotypic cultures were raised to the air/liquid interface to induce stratification and differentiation for 6 days. The regenerated tissue was then embedded OCT before cryosectioning and imaging. The TUNEL assay of CTRL and CPSF1 siRNA tissues was performed by In Situ Cell Death Detection Kit, TMR red (Roche) following the manufacturer's instruction, and cells with positive signals were counted using Image J software (NIH). In the progenitor competition assay, keratinocytes were labeled by infection retrovirus expressing H2B-GFP or H2B-mCherry before the nucleofection of CTRL or CPSF1 siRNA. For quantification, the GFP or mCherry expressing keratinocytes in the regenerated epidermis were counted using the Image J software (NIH). For each image, quantification was performed for the entire thickness of epidermal tissue as well as the top and bottom halves of the tissue divided evenly using Image J.

**Nile Red staining.** Nile Red staining of tissue sections was performed similar to the method previously described[57], with our experimental details listed below. The stock solution was prepared by dissolving Nile Red (Sigma 72485) first in acetone at 500 μg/mL then diluting to a working solution of 2.5ug/mL. Skin tissue sections were fixed in 10% formalin for 5 min, rinsed briefly in PBS, then incubated in the working solution for 10 min at room temperature. The slides were subsequently washed with PBS and stained by NucBlue Fixed Cell ReadyProbes Reagent (DAPI, Invitrogen) for 5 min, and were mounted with anti-Fade Fluorescence Mounting Medium (Abcam). For quantification, the thickness of Nile-Red-positive regions in each image was measured at 3 regions using the Image J software (NIH). The average thickness from 3 measures of each image were calculated, and a total of 26 images per condition were included for statistical analysis (T-test) using Prism.

**CPSF1 complex purification and protein identification using mass spectrometry.** Keratinocytes were trypsinized, washed in PBS, and resuspended in 200 μL hypotonic buffer (10 mM HEPES at pH 7.4, 1.5 mM MgCl2, 10 mM KCl, 1× protease inhibitor cocktail (Roche)) per million cells. Cells were lysed by adding an equal volume of hypotonic buffer with 0.4% NP-40 for 2 min. Nuclei were pelleted by centrifugation at 4000 rpm and lysed in ten cell pellet volumes of nucleus lysis buffer (50 mM Tris at pH 8.0, 0.05% igepal, 10% glycerol, 2 mM MgCl2, 250 mM NaCl, protease inhibitor cocktail (Roche)). Nuclei were sheared with a 27.5-gauge needle, and lysis proceeded for 30 min. Insoluble material was removed by centrifugation at 13,000 rpm for 10 min, and nuclear supernatant was used for purification. Dynabeads™ Protein G (Thermo Fisher) were conjugated with CPSF1 mouse monoclonal antibody (Santa Cruz Biotechnology, Inc.) or mouse IgG as control, added to nuclear supernatant for 4 °C incubation overnight, and washed five times with nucleus lysis buffer. Proteins were boiled off from beads and separated on SDS-PAGE for immunoblotting or mass spectrometry identification. For protein identification, immunoprecipitations were first separated on SDS-PAGE and stained with colloidal blue (Life Technologies). Gel slices (0.5 cm) were submitted to the Northwestern Proteomics Facility for mass spectrometry analyses.

**RNA-seq.** RNA was extracted using Quick-RNA™ MiniPrep (Zymo Research) with DNase I treatment. RNA-seq libraries were prepared using NEBNext Ultra™ Directional RNA Library Prep Kit for Illumina (New England BioLabs) with ribosomal-RNA depletion (New England BioLabs) or NEBNext® Poly(A) mRNA Magnetic Isolation Module (New England BioLabs). Libraries were sequenced as single-end 50-base-pair (bp) reads using the Illumina HiSeq 4000 platform by Northwestern University NUSeq Core facility.

**3′READS +library construction.** 3′READS+ experiments were performed as described[34] with minor modifications to enable multiplexing on the Illumina HiSeq 4000 platform, with our experimental details listed below. Poly(A)+ RNA from keratinocytes was captured from 15 μg total RNA using NEBNext® Poly(A) mRNA Magnetic Isolation Module (New England BioLabs). Fragmentation was performed on the beads using ShortCut® RNase III (New England BioLabs). The fragmented poly(A)+ RNA was ligated to 5′ adapter (5′ -CAGACGUGUGCUCUUCCGAUC UNNNN) on the beads with T4 RNA ligase I (New England BioLabs). The ligation products were captured and poly(A)-tail-trimmed by RNase H (New England BioLabs) on biotin-T15-(+TT)5 (Exiqon) bound to Dynabeads MyOne Streptavidin C1 (Thermo Fisher). The RNA fragments were then ligated to 3′ adapter (5′ -rApp/NNNN AGATCGGAAGAGCGTCGTGTAG/3ddC) with T4 RNA ligase 2, truncated KQ (New England BioLabs). The ligation products were then reverse transcribed by M-MLV reverse transcriptase (Promega), followed by PCR using Phusion high-fidelity DNA polymerase (Thermo Fisher) and NEBNext® Multiplex Oligos for Illumina® (New England BioLabs) for 15 cycles. PCR products were size-selected with AMPure XP beads (Beckman Coulter), and sequenced by Illumina HiSeq 4000 platform with 1x50bp by the NUSeq Core facility at Northwestern University.

**qRT-PCR expression analysis.** Total RNA was extracted using Quick-RNA™ MiniPrep (Zymo Research), and reverse transcribed using the SuperScript VILO cDNA synthesis kit (Invitrogen). For quantification of the proximal/distal PolyA-site usage ratio, an additional step of poly(A)+ RNA isolation was performed using NEBNext® Poly(A) mRNA Magnetic Isolation Module (New England BioLabs). qPCR was performed using the SYBR Green Master Mix (Thermo Fisher) or EvaGreen Master Mix (Bullseye). Samples were run in duplicates and normalized to levels of 18S ribosomal RNA for each reaction. Statistical analysis such as one-way ANOVA and Two-tailed Student's unpaired t-test was calculated using GraphPad Prism7. Bar graphs and their associated error bars are represented as mean ± standard deviation. Primer sequences used for qPCR are listed in Table S5.

**RNA-seq analysis.** RNA-seq libraries were aligned to the hg19 genome using Hisat2[58]. Browser tracks were generated using the UCSC genome browser. Htseq[59]-count was used to generate counts tables at all genes. Differential expression analysis was done using DESeq2[60]. Genome browser tracks of RNA-seq were normalized by sequencing depth.

**3′READS+ analysis.** 3′READS+ data analysis was performed based on the protocol described in Zheng et al.[34]. 5′ adapters were removed using CutAdapt, followed by removal of 3′ adapter consisting of four random nucleotides. T's corresponding to PolyA tails were removed from reads and saved. Up to one non-T base was allowed in T tails. Reads < 22 nucleotides were filtered out, and the remaining reads were aligned using bowtie2 in end-to-end mode. Aligned reads with MAPQ score less than 10 were filtered out. Sequence through 20 bp downstream of PolyA site was obtained. If the T tail of a read had a non-T base, and the sequence from the start of the T tail to the non-T base aligned to the genome, then that portion of the T tail was removed and considered part of the aligned sequence. Reads in which the T tail had at least two unmappable T's were considered Poly-A Site supporting (PASS) reads. Poly-A sites from PASS reads were merged by 24 nucleotides. Sites corresponding to hg19 blacklist version 2 (Boyle lab), retrotransposons (ucscRetroInfo5 table), microRNA, and snoRNA (ucsc hg19 refseq

table) were filtered out. We then filtered out poly-A sites that did not overlap with a gene according to the GENCODE version 17 annotation from UCSC. A table of counts was built at the remaining sites for each library.

To identify differential usage of Poly-A sites, the sum of counts at each gene was calculated. The relative "Fraction of PolyA site Usage" (FPU) was then calculated as counts at each site divided by the total counts at the gene associated with that site. EdgeR was used to calculate differential usage and p-value based on the FPU.

Several filters were applied to 3′READS+ PolyA site analyses to assure we focused on stably expressed genes and functional PolyA sites. PolyA sites met the following criteria (1) gene expressed in keratinocytes (at least 5 counts in either UD or DF RNA-seq libraries) (2) the sum of the counts from the two biological replicates ≥ 10, and (3) FPU from at least one library ≥10%.

The 18173 remaining sites were annotated using GENCODE version 17. 1118 sites overlapped with multiple genes and were removed from further analysis. We designated Poly-A sites as Introns, UTRs, or Exons using bed files for those regions from GENCODE version 17. We prioritized introns to ensure that previously uncharacterized intronic sites would not be incorrectly annotated as UTRs. 4885 sites were designated as introns. From the remaining sites, 11734 were identified as UTRs. The final 436 sites were annotated as exons.

To correct for potentially incorrect annotations of UTR's as introns, we reasoned that sites associated with genes that only appeared once should be designated as UTRs. We identified 4525 of these sites. While 3566 of these were already correctly annotated as UTR's, we also adjusted the annotation of the remaining 959 sites (54 exons, 905 introns) to UTR. Given that GENCODE tables used for annotation included regions previously filtered out, one final step was required to refine our list. From the annotated list of 17,055 sites, 2429 were ambiguous as they are located in overlapping genes. Even if one of these genes did not meet the filters previously stated, it was impossible to distinguish whether the reads corresponded to the filtered out gene or not. Removal of these sites left us with a robust final list of 14625 poly-A sites.

To determine the 428 differentially used Intronic PolyA sites in Fig. 1a, we used a fold-change cut off of 2. Sites were validated using RNA-seq data. We combined three replicates of undifferentiated (UD) and three replicates of differentiated (DF) keratinocytes and built a table of counts from RNA-seq at PolyA site ±100 bp and calculated log2FC from UD to DF conditions. Sites with fold change >1.5 in the same direction in both 3′READS+ libraries and RNA-seq libraries and with more counts upstream than downstream the PolyA site, were considered alternatively used IpA sites. To identify which of these sites were regulated by CPSF, we built a table of counts from CPSF1-knockdown and control 3′READS+ data at PolyA sites and calculated FC. Sites with FC 1.5 that changed in the same direction as in UD DF 3′READS+ data were considered regulated by CPSF. The height of 3′READS+ genome browser tracks was normalized based on sequencing depth.

For motif analysis of the PolyA sites, DNA sequences were extracted with an extension of 200 bp both upstream and downstream. MEME[61,62] motif search was performed using the RNA (DNA-encoded) motif database "Ray2013 Homo sapiens (DNA-encoded)", using a background model of 1-order that adjusts for dimer biases.

**Reporting summary**. Further information on research design is available in the Nature Research Reporting Summary linked to this article.

## Data availability

The authors declare that all data supporting the findings of this study are available within the article and its supplementary information files or from the corresponding author upon reasonable request. Raw and analyzed 3′READS+ and RNA-seq data generated by this study have been deposited in the GEO database under the accession code: GSE127223. Other raw data are included in the "Source Data" file. Other publicly available genomic data sets used in this study include: GSE111310 (IpA sites in immune cells), GSE32883 (MYC ChIP-seq), GSE33480 (HCT116 RNA-seq). Source data are provided with this paper.

## Code availability

The authors used for 3′READS processing are available on Github at: https://github.com/Bao-Lab/3READS. These scripts are accessible with no restriction.

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

## Acknowledgements

This work is supported by a NIH K99/R00 Award (R00AR065480), a NIH R01 (AR07515), the Searle Leadership Fund, the Northwestern Skin Disease Research Center Pilot & Feasibility Award, the Basic Insights Award from Northwestern Cancer Center to X. B., as well as a NIH CMBD training grant (T32GM008061) and a Northwestern Presidential Fellowship to S.M.L. We appreciate the support from the Skin Biology and Diseases Resource-based Center (SBDRC, P30AR075049) for providing tissues and culture media for this study. We would like to thank Y. Yu for cloning and testing shRNAs during his rotation, and we are grateful for the input from all other lab members.

## Author contributions

X.C. performed most of the experiments. S.M.L played a leading role in data analysis. J.K. contributed to immunostaining, cloning, RNA-seq library construction for this project. G.M.G. contributed to cloning of the constructs generated from this study. X.B., X.C., and S.M.L designed the study and wrote the manuscript with approval from all the authors.

## Competing interests

The authors declare no competing interests.
