## [Peer Review File · Nature Communications]

Reviewers' Comments:

Reviewer #1:

Remarks to the Author:

The current manuscript by Chen et al., investigated the role of CPSF-HNRNPA3-dependent polyadenylation in the regulation of epidermal progenitor differentiation via GRHL3. GRHL3 is known to regulate not only epidermal differentiation but also epidermal barrier repair, barrier function and wound repair. What is the role of intronic polyadenylation in these homeostatic processes? What are the consequences of premature terminal differentiation following RNAi/KO of CPSF?

The authors have followed the same logic described in their previous findings on epidermal progenitor differentiation (Refs 2 and 10). They showed a GRHL3-dependent mechanism without validating their findings through an epidermal functional assay. Can the authors exploit CPSF-HNRNPA3 to enhance wound repair or barrier function in barrier impairment settings? or to prevent skin cancer in a Grhl3-related model, etc.

Minor comments:

- The authors assessed progenitor differentiation of the IFE cell population. Can their findings be applied to progenitors of other skin compartments?
- The majority of sites are used in UD (66%) compared to differentiated keratinocytes with no consideration of the spontaneous differentiation in cells that are considered UD. Can the authors sort and culture basal stem cells (UD) to compare with suprabasal differentiated cells and to validate their IpA sites (Fig.1)? Are there any truncated proteins generated by IpA that may influence progenitor differentiation? How specific is the CPSF overexpression to GRHL3 in progenitors versus stem cells?
- Again in Fig.2, out of the 621 genes regulated, the authors choose to manipulate GRHL3 without sufficient justification. Given the strong phenotype of Fig.2f, how much of this solely account to GRHL3 and not to other differentiation factors or EDC? What is the differentiation status of CPSF CRISPRi cells with GRHL3 RNAi? Does the function of the skin reflect the changes in gene expression?
- Which Grhl3 isoform is affected? Can the authors show a Grhl3 western blot?
- The authors show increased S100A8/A9 that is associated with differentiation. This does not correlate with previous studies showing S100A8/A9 upregulation in Grhl3-deficient keratinocytes. Which direct target genes are regulated downstream of GRHL3?
- What is the consequence of single HNRNPA3 knock-down on epidermal differentiation?
- Are there any IpA sites in Exo59 or PRMT1 that would lead to indirectly regulating GRHL3 levels? Can HNRNPA3 modulate PRMT1 to affect GRHL3 levels, as previously published by the authors?

The authors did not perform any skin function assay following CPSF loss and premature differentiation.

CPSF polyadenylation was previously shown to regulate cell differentiation (Singh et al., 2018 Nat Commun), and this is well established for GRHL3 in keratinocyte progenitors. The Exo59 and PRMT1 papers (Refs 2 and 10) are to some extent similar skin differentiation stories, showing a GRHL3-dependent differentiation mechanism. Therefore, this manuscript lacks novelty.

Reviewer #2:

Remarks to the Author:

In this manuscript, Chen and colleagues studied differential intronic polyadenylation (IpA) during human keratinocyte differentiation. They further identified a prominent IpA site within the first intron of GRHL3, an important TF for epidermal differentiation, and demonstrated the downregulation of CPSF or the deletion of the GRHL3 IpA site increased the expression of full-length GRHL3 in progenitor keratinocytes. They further identified a number of proteins interacting

with CPSF including hnRNPA3, which plays a role in alternative splicing. They propose a mechanism by which CPSF and hnRNPA3 function to reduce the production of full-length GRHL3 mRNA in progenitor keratinocytes by promoting alternative splicing and intronic polyadenylation within the first intron. Overall, this is an interesting study which provides new insights into how differential IpA sites usage plays a role in epidermal differentiation. That being said, many more analysis and detailed studies should be carried out to strengthen the study in the potential revision.

I have the following comments:

1. In Fig 1, they performed 3'READS+, an improved 3'end sequencing technique, with undifferentiated and differentiated human keratinocytes. Although they mostly focus on IpA sites, they should also identify differentially used polyA sites within the 3'UTRs and determined 1) how many genes show differentially used polyA sites in canonical 3'UTRs; 2) whether the use of proximal sites is a feature for epidermal differentiation.
2. Among 2703 IpA sites, they identified 610 differentially used sites. Motif analysis should be performed to determine whether these 610 sites have unique features that promote the differential usage (over- or under-used during differentiation), in comparison to the unchanged IpA sites as well as to changed and unchanged canonical 3'UTR sites.
3. Similarly, they found 379 sites that are shared between keratinocytes and immune system. They should determine 1) whether the same host gene used different or identical IpA sites; 2) whether the small overlap is due to the differential expression of the host genes or the differential usage of IpA sites even when the same host genes are expressed. If the latter case is true, do the differentially used IpA in keratinocyte vs immune systems have different motifs?
4. They found 66.4% of IpA sites are preferentially used in undifferentiated cells and they showed a few examples including ESPN and IQCK. However, is there any functional relevance of these 405 genes during epidermal differentiation? Do ESPN and IQCK play any role in epidermal differentiation? In other words, is there any global impact of differential IpA usage other than the case of GRHL3?
5. All Western results in Figs. 1-3, 6 should be repeated and quantified.
6. In Fig. 2 studies, they knocked down CPSF1, observed downregulation of multiple CPSF subunits at the protein level and detected increased expression of epidermal expression. To confirm the changes of other CPSF subunits are at the protein level, they should also measure mRNA by qPCR. They nicely showed the difference in epidermal differentiation in an organotypic system when knocking down CPSF. However, more thorough analysis should be performed in this system such as whether KD CPSF leads to the reduction or deletion of basal stem cells, whether basement membrane is affected and whether proliferation/cell death is affected.
7. They used both RNAi and CRISPRi to repress CPSF1 to reduce off-target effect. A head-to-head comparison of RNA-seq datasets derived from both inhibitory studies should provide the most critical insights into consistency and identify commonly altered genes independent of experimental approaches to inhibit CPSF.
8. It is unclear why the H73A CPSF3 mutant can function as dominant negative. Does the mutant bind to nascent RNA but fail to process them? If that's the case, could they observe many mRNAs w/o polyA tail and can these abnormal mRNA be stabilized? It seems this approach will interfere with normal mRNA metabolism rather than inhibiting IpA alone. Unless they can provide more robust characterization such as 3'READS+ results, I am not convinced that these data will provide a clear argument for their role in suppressing premature differentiation.
9. They showed that 399 out of 610 differentially used IpA sites during epidermal differentiation is

also affected by the downregulation of CPSF. This suggests that these sites are more sensitive to the levels of CPSF than the remaining 211 sites. They should perform motif analysis to determine if these sites harbor some unique sequences. In addition, they should also analyze whether mRNA levels are higher or lower for these 399 sites when comparing to the unchanged sites.

10. In Fig. 5, they studied the GRHL3 site in more details with CRISPR deletion. However, I'm not sure whether Fig. 5b clearly demonstrated the deletion pattern. They should use either deep sequencing or at least sequence 10~20 clones of the DNA fragment rather than using the mixture. More importantly, they should try to derive single clones from these CRISPR experiments if possible. They could have a much better understanding of how the deletion of this site affects GRHL3 expression and keratinocyte in general. In its current form, they only have a mixture of cells with ~50% deletion efficiency, and this doesn't count for heterozygous deletion events. To show the increase of GRHL3, they should perform Western quantification at minimum.

11. In Fig. 7, they aimed to identify CPSF-interacting proteins, whose KD synergize the control of GRHL3 IpA. They identified hnRNPA3 as a candidate. They went on to show that hnRNPA3 may affect an alternative splicing event within the same intron of GRHL3. However, it is not clear to me why the alternative splicing could further enhance the IpA usage by GRHL3. Does the recruitment of hnRNPA3 to the intron directly increase the binding of CPSF to the IpA or the splicing itself recruits CPSF? Despite the provided evidence, it remains how the synergy between hnRNPA3 and CPSF is generated. Regardless the potential mechanism, I am also curious about whether the hnRNPA3/CPSF duo only functions on the GRHL3 site or other sites may also be affected.

Reviewer #3:

Remarks to the Author:

In the manuscript by Chen et al, the authors explore the mechanism of keratinocyte differentiation and the role that the CPSF complex plays in regulating altered cleavage and polyadenylation. The authors first utilize a well-characterized primary keratinocyte human cell line and perform both RNA-seq and 3READS+ both before (UD) and after (DF) differentiation. They observe significant reduction in the usage of intronic poly(A) sites (IpA) sites as cells differentiate. They then determine that members of the core CPSF complex are being downregulated as cells differentiate. Using two distinct approaches (RNAi and CRISPRi), they then determine that downregulation of CPSF1 alone is sufficient to stimulate differentiation and drive down usage of IpA sites. They then use CRISPR to drill down on the importance of a specific IpA site within GRHL3 and show that deletion/mutation of this site increases its expression. Finally, they conduct IP MS/MS analysis of CPSF complexes before and after differentiation to identify interacting proteins that are specific to the UD state and focus on one of those factors, hnRNPA3, to determine its functional relevance. Overall, this is a nice study that has many novel findings. In fact, I have just a series of minor concerns for Figures 1-5.

Having said this, I have issues with the data presented in Figures 6 and 7. It almost seems that these two figures could be removed and the overall strength of the study goes up. My first issue is the simple logic of conducting the mass spec. The authors state that 'differential levels of CPSF proteins...raises the possibility of differential protein interactions in the two conditions...why, this doesn't seem to be logical or at least the logic needs to be presented better. Even more confusing, the authors show that many of these 'CIPs' are also downregulated just like the CPSF proteins. So, ultimately, the levels of CPSFs and CIPs go down together making it strange to me that a differential association would occur.

To begin with, the methods suggest that the mass spec was not done using RNAse treatment and it should. One of the supplemental data uses RNAse in the validation but the initial mass spec should be. The authors confirm mass spec association using specific IPs but they appear to be only in the UD cell lysates. Shouldn't the authors validate the findings of differential association so IPs from both UD and DF be done in comparison? They focus on hnRNPA3 but the experiments raise

more questions than they answer. The authors need to do both RNA-seq to address splicing changes and 3'READS+ once hnRNPA3 is knocked down. Finally, it is this reviewer's opinion that the authors left the most interesting question on the table: How does the CPSF complex get downregulated?? Unfortunately, there is no data presented on this question.

Other issues:

- 1) The 232989 polyA sites were distributed over how many genes?
- 2) The fact that ~10% of all poly(A) sites are intronic poly(A) sites coupled with the fact that very few overlap with other recently published data is potentially concerning. Can you authors provide evidence that internal priming isn't happening or priming to degraded RNA that has been adenylated by the TRAMP complex? Can you authors comment on what percentage of their IPAs have already been annotated? Can the authors also do a MEME analysis to provide evidence as to what percentage of the IPA sites contain polyA motifs upstream? This also would provide additional confidence that the sheer number of IPA sites is a real thing.
- 3) In figure 1D, it seems to be a peculiar choice to measure CPSF members in comparison to Rpb1. The authors should also consider measuring members of the CstF, CFIm, and CFIIIm complex as these are also considered to be the major players in cleavage and polyadenylation.
- 4) Figure 1j is not overly convincing. The authors should consider additional images to convey their main point.
- 5) Figure 6F, the authors should label or at least mention in the legend what percentage of input and pulldown are loaded on the gels

Minor

- 1) Typo 'catalytic-inactivate' should be '-inactive'

• Point by Point Response to Reviewer #1

We thank the feedback from Reviewer1. In particular, we appreciate the suggestions to broaden the scope and to examine other processes in addition to keratinocyte differentiation. The details are listed below. (*Reviewer's original comments are in Gray, responses are in Black, additions of new data are indicated in Blue*).

The current manuscript by Chen et al., investigated the role of CPSF-HNRNPA3-dependent polyadenylation in the regulation of epidermal progenitor differentiation via GRHL3. GRHL3 is known to regulate not only epidermal differentiation but also epidermal barrier repair, barrier function and wound repair. What is the role of intronic polyadenylation in these homeostatic processes? What are the consequences of premature terminal differentiation following RNAi/KO of CPSF? The authors have followed the same logic described in their previous findings on epidermal progenitor differentiation (Refs 2 and 10). They showed a GRHL3-dependent mechanism without validating their findings through an epidermal functional assay. Can the authors exploit CPSF-HNRNPA3 to enhance wound repair or barrier function in barrier impairment settings? or to prevent skin cancer in a Grhl3-related model, etc.

We agree with the reviewer that it is important to examine the roles of CPSF and IpA in the processes other than keratinocyte differentiation. We have performed several experiments and analyses to address this concern.

1) To determine the roles of IpA in keratinocyte migration, we generated migrating keratinocytes using methods as reported previously¹. In brief, a grid of scratches were made on a monolayer of keratinocytes. IpA usage was quantified for keratinocytes with or without scratches. As compared to the drastic changes of IpA in differentiation, wounding and migration minimally affected the usage of these sites (*new data included as Supplementary Fig. 4b*). For example, the usage of GRHL3 IpA between undifferentiated and differentiated keratinocytes differs > 100 fold, yet only ~1.18 fold difference was observed in migrating keratinocytes versus control. The IpA sites in ALOX15B and CRBN, which were drastically downregulated in keratinocyte differentiation, were not significantly altered in keratinocyte migration. Thus the differential usage of IpA sites we uncovered in this study, including GRHL3 IpA, is more relevant to the process of differentiation than migration.

2) We examined the roles of CPSF in keratinocyte migration. Interestingly, we found that CPSF knockdown significantly impaired cell migration in scratch assay (*new data included as Fig. S2g, h*). Thus CPSF impacts multiple functions that are crucial for the physiological functions of keratinocytes, including both differentiation and migration. A total of 737 CPSF target genes, shared by both RNAi and CRISPRi, were identified in this study. The top 5 most enriched Gene Ontology (GO) terms of upregulated genes are related to "keratinocyte differentiation". Other GO terms, although less enriched, were also associated with CPSF knockdown. Related to this migration process, "negative regulation of cell migration" is also a GO term associated with genes upregulated with CPSF knockdown. (*The full list of GO terms are now included in Table S3.*) These genes include *CYP1B1*, *ALOX15B*, *KRT16*, *SERPINE1*, and *TP53INP1*. These findings suggest that CPSF promotes keratinocyte migration through suppressing several negative regulators of migration.

GRHL3 is upregulated with CPSF knockdown. As pointed out by the reviewer, GRHL3 is important for both differentiation and migration. Differentiation and migration are two distinct functional states of keratinocytes, which involve GRHL3 binding at different genomic locations to control different sets of target genes¹. GRHL3's role in differentiation or migration is highly dependent on the context. GRHL3 is moderately expressed in undifferentiated keratinocytes, and GRHL3 is upregulated in keratinocyte differentiation. It had been previously demonstrated that GRHL3 overexpression (gain-of-function) was sufficient to promote differentiation marker gene expression in keratinocytes cultured in undifferentiation condition²; GRHL3 knockdown (loss-of-function) in undifferentiated keratinocytes impaired migration¹. In the case of CPSF knockdown or GRHL3 IpA KO, GRHL3 is upregulated in keratinocytes cultured in undifferentiation condition. In agreement with the findings in the literature, we found that this upregulation of GRHL3 by CPSF and IpA is functionally involved in promoting differentiation.

3) We uncovered that the maintenance of CPSF expression in undifferentiated keratinocytes requires the proto-oncogene MYC. MYC directly binds to the promoter region of CPSF and is required to sustain CPSF expression in undifferentiated keratinocytes (*new data included as Fig. 4*). MYC knockdown also strongly influences IpA (including GRHL3 IpA), similar to CPSF knockdown (*new data included as Fig. 5j*). These findings place both CPSF and IpA in a broader regulatory network downstream to MYC.

[Redacted]

These new data, generated thanks to the suggestions from the reviewer, highlighted the importance of CPSF in both keratinocyte differentiation and migration, placed the function of CPSF and IpA in a broader regulatory network, and opened very interesting avenues for future studies.

To better evaluate the consequences of CPSF knockdown, we performed a “progenitor competition assay” to quantify the regenerative capacity of keratinocytes (new data included as Fig. 2b-c). In this technically improved version of “progenitor competition assay”, epidermal tissue was regenerated using 50% keratinocytes labeled by H2B-GFP (Green) and 50% keratinocyte labeled by H2B-mCherry (Red). Expression of fluorescently-tagged H2B enables specific labeling of the nuclei, instead of diffusing fluorescence across the entire cells. This improved strategy facilitates software-assisted quantification using ImageJ. When Red or Green labelled keratinocytes were both treated with non-targeting-control siRNA, the regenerated tissue using these cells showed a comparable amount of red versus green labeled cells. When Red keratinocytes were treated with CPSFi and Green keratinocytes were treated control siRNA, the regenerated tissue showed significantly reduced Red versus Green ratio ($p < 0.0001$). These data indicate that CPSF loss in keratinocytes strongly impaired their regenerative capacity.

We appreciate the suggestion of evaluating the barrier function of regenerated epidermal tissue, comparing CPSF knockdown versus control. Several barrier function assays, such as the assays using β -gal or toluidine blue, are well established for mouse models. These assays are not well-suited for organotypically regenerated epidermal tissue. The pieces of regenerated tissue have very small surface area (1cmx1cm), with the dermal scaffold openly exposed on the edges and from the bottom. Dye can penetrate easily from the open dermal areas, interfering with the assessment of epidermal barrier function. After a few trial and error as well as careful consideration, we selected Nile Red staining of tissue sections as an alternative approach. Nile Red stains neutral lipids, a critical component that supports the epidermal barrier function to prevent water loss. Software-aided quantification, using ImageJ, allows efficient processing of dozens of images to obtain statistical power. Overall, we found that CPSF knockdown led to significantly increased thickness of the lipid-rich layer, suggesting increased barrier function for preventing water-loss (new data included as Supplementary Fig. 2e,f).

Minor comments:

- *The authors assessed progenitor differentiation of the IFE cell population. Can their findings be applied to progenitors of other skin compartments?*

We appreciate this very interesting question. In addition to IFE cell population, skin has multiple other cell types including melanocytes, fibroblasts, as well as the immune cells. Between keratinocytes and the immune cells, we found that 17% of the IpA sites identified in keratinocytes are shared with the IpA sites found in the immune cells. Interestingly, when we compared the genes associated with these IpA sites between keratinocytes and the immune system, we found that the degree of overlap increased from 17% to 45.3% (new data included as Supplementary Fig. 1d). For example, 3 IpA sites in AGO3 were found in keratinocytes, and 4 IpA sites in AGO3 were found in the immune cells. Only 2 of these overlap. These data suggest that different IpA sites are used even within the same genes in different cell types. Thus the different cell types in skin are likely to use different spectrums of IpA sites.

For GRHL3 IpA, analysis of the RNA-seq data from the ENCODE project revealed that this GRHL3 IpA site is also used in HCT116 cells. CRISPR KO of this IpA in HCT116 cells, using both bulk cell analysis as well as single-clone characterization, also increased GRHL3 mRNA expression (new data included as Supplementary Fig. 5c-h). These findings indicate that the mechanism of using the IpA site to suppress GRHL3 expression is not limited to human keratinocytes.

- *The majority of sites are used in UD (66%) compared to differentiated keratinocytes with no consideration of the spontaneous differentiation in cells that are considered UD. Can the authors sort and culture basal stem cells (UD) to compare with suprabasal differentiated cells and to validate their IpA sites (Fig.1)? Are there any truncated proteins generated by IpA that may influence progenitor differentiation? How specific is the CPSF overexpression to GRHL3 in progenitors versus stem cells?*

We generally only uses early-passage keratinocytes for our experiments. To determine if spontaneous differentiation also impacts CPSF expression and IpA usage, we generated late-passage (p6) keratinocytes by

culturing early-passage (p2) for 20 additional days. These late-passage keratinocytes show significant upregulation of differentiation markers (p16, S100A8) and downregulation of proliferation markers (Ki67, AURKB). The expression of genes encoding CPSF subunits were not drastically changed between late- versus early-passage cells. Specifically, 4 out of the 6 genes encoding the CPSF complex, were not significantly changed. 2 out of the 6 genes were decreased to ~50% in late-passage keratinocytes as compared to early-passage keratinocytes (new data included as Supplementary Fig. 3b,c). In the case of calcium-induced keratinocyte differentiation, all 6 CPSF subunits are reduced to ~ 30% or less. We also quantified IpA usage between early- versus late-passage keratinocytes. Similar to the trend of CPSF expression, the usage of representative IpA sites was mildly reduced. [Redacted]

Thus culture-induced differentiation has very limited impact on CPSF expression and IpA usage, as compared to the drastic changes in calcium-induced differentiation.

We agree with the reviewer that it is both interesting and important to test potential protein truncation resulted from IpA. A previous study validated several cancer-specific protein truncations resulted from IpA³. These IpA sites are not used in normal cells. In the case of GRHL3 IpA in keratinocytes, it occurs within the first intron. The corresponding mRNA would only translate the first 6 amino acids of the protein. We also surveyed a number of other genes with IpA sites. There are two bottlenecks for this direction. First, an antibody must be raised against the N-terminal regions of target proteins, as IpA would truncate the c-terminal regions of the proteins; Second, the antibody needs to be of high-quality, with low background and high sensitivity to detect the truncated protein. We have screened a total of 8 different commercial antibodies (details included in Table R1 in Appendix II of this letter) and have not yet identified an antibody that is good enough to clearly identify potential truncated proteins from other IpA targets.

In terms of CPSF expression in progenitors versus stem cells, a recent paper reported that CPSF expression is increased in iPS cells as compared to a variety of embryonic tissues⁴. Thus high CPSF expression is not restricted to progenitors and is likely to play essential roles in stem cell function.

- Again in Fig.2, out of the 621 genes regulated, the authors choose to manipulate GRHL3 without sufficient justification. Given the strong phenotype of Fig.2f, how much of this solely account to GRHL3 and not to other differentiation factors or EDC? What is the differentiation status of CPSF CRISPRi cells with GRHL3 RNAi? Does the function of the skin reflect the changes in gene expression?

We appreciate this great suggestion to clarify the rationale for choosing to focus on GRHL3. The manuscript has been updated to better reflect our rationale. In brief, the intersection of genes with CPSF-dependent IpA sites and CPSF RNA-seq targets revealed a total of 14 genes. Remarkably, the 3'READS+ data and RNA-seq data of these genes showed anti-correlation (new data included as Fig. 5a,b), which suggests a possibility that the usage of the IpA sites may suppress gene expression. Among these genes, GRHL3 IpA shows the highest level of FPU (fraction of PolyA site usage) in undifferentiated keratinocytes, suggesting this site may have a strong impact on the total mRNA expression.

In the case of CPSF CRISPRi cells with GRHL3 RNAi, a total of 6 differentiation genes were quantified. All 6 genes were upregulated by CPSF CRISPRi alone. With the combination of GRHL3 RNAi and CPSF CRISPRi (double knockdown), 5 of these 6 were restored to the level comparable to control. These data suggest that the upregulation of these genes in CPSF CRISPRi is mediated by GRHL3. In the case of SPRR1B, double knockdown reduced the upregulation from ~5 fold to ~ 3 fold. Thus GRHL3 knockdown was not able to rescue 100% of the differentiation gene upregulation in CPSF knockdown, and GRHL3 upregulation is not the only mechanism contributing to the differentiation phenotype induced by CPSF knockdown. These keratinocytes with double knockdown were still partially differentiated.

- Which Grhl3 isoform is affected? Can the authors show a Grhl3 western blot?

For this study, we generated strand-specific RNA-seq data. The strand specificity of our data provides the capacity to better identify the isoforms, as GRHL3 overlaps with the neighboring gene STPG1 on the opposite strand. Our RNA-seq data clarified that the isoform GRHL3 (NM_198173) is predominantly expressed in keratinocyte differentiation (The RNA-seq tracks are now included as Supplementary Fig. 5c). [Redacted]

We agree with the reviewer that it is important to validate GRHL3 protein using western blotting. A previously validated rabbit antibody⁵ is now discontinued from the vendor. We purchased and tested a total of 4 other antibodies raised against GRHL3, which are currently commercially available. As GRHL3 is strongly upregulated during keratinocyte differentiation, our first step of validation was to compare the pattern between undifferentiated and differentiated keratinocyte lysate. (According to our RNA-seq, GRHL3 is upregulated ~30 fold in differentiation.) Only 2 of these 4 antibodies showed an increased band corresponding to the size of GRHL3 in the differentiation condition. The level of GRHL3 in undifferentiated keratinocyte is below the sensitivity of the detection, with a number of non-specific bands. In the heterogeneous population of keratinocytes with GRHL3 KO (~4-5 fold upregulation, a lot less than differentiation), we could see a very weak band appearing yet the data is not of publication-quality. We have also tried the HCT116 KO line. GRHL3 expression is lower in HCT116 cells than in keratinocytes, and the GRHL3 band was also below the detection sensitivity limit of these antibodies. The quality and sensitivity of commercial antibodies are a limitation for this test. The list of tested antibodies are included in Table R2 in Appendix II of this letter.

- The authors show increased S100A8/A9 that is associated with differentiation. This does not correlate with previous studies showing S100A8/A9 upregulation in Grhl3-deficient keratinocytes. Which direct target genes are regulated downstream of GRHL3?

We appreciate this very interesting point raised by the reviewer. S100A8 and S100A9 are implicated in both keratinocyte differentiation as well as inflammatory responses.

Both S100A8 and S100A9 are strongly upregulated during keratinocyte differentiation^{2,6,7}. In keratinocytes cultured in differentiation condition, GRHL3 knockdown impaired the expression of S100A8 and S100A9⁸. This findings indicate that S100A8 and S100A9 are downstream to GRHL3 during keratinocyte differentiation. Our findings that GRHL3 knockdown downregulated S100A8 and S100A9 in CPSF knockdown agree with the previous findings in the differentiation state.

We expanded our qPCR validation to include additional GRHL3 targets SBSN and CRCT1. These two genes were validated as direct GRHL3 targets⁸, although the raw data of GRHL3 ChIP-seq in differentiation condition is not publicly available. Both genes were upregulated with CPSF knockdown, and GRHL3-CPSF double knockdown restored SBSN and CRCT1 expression. These genes were also significantly altered with GRHL3 IpA KO (new data included as Fig. 5e,i,j).

In addition to their roles in keratinocyte differentiation, both S100A8 and S100A9 are involved in inflammation and wound healing⁹. GRHL3 ChIP-seq data generated using migrating keratinocytes are publicly available¹. These data indicate that GRHL3 is not enriched at the promoter regions of S100A8 and S100A9, suggesting that S100A8 and S100A9 are likely to be indirectly induced by GRHL3 loss in keratinocytes cultured in undifferentiation condition.

- What is the consequence of single HNRNPA3 knock-down on epidermal differentiation?

To determine the impact of HNRNPA3 on gene regulation in undifferentiated keratinocytes, we generated RNA-seq data to compare HNRNPA3 knockdown versus control. In total, we identified 1490 genes that are significantly altered with HNRNPA3 knockdown. 69% of these genes overlap with calcium-induced differentiation signature, although only 20.5% of these genes overlap with CPSF target genes. These data indicate HNRNPA3 has both CPSF-dependent and CPSF-independent roles in controlling keratinocyte differentiation. These new data are now included in Fig. 7a-d.

- Are there any IpA sites in Exo59 or PRMT1 that would lead to indirectly regulating GRHL3 levels? Can HNRNPA3 modulate PRMT1 to affect GRHL3 levels, as previously published by the authors?

We appreciate these suggestions to explore connections of our findings with 2 previous studies. According to our 3'READS+ data, neither EXO59 nor PRMT1 has IpA. HNRNPA3 knockdown only mildly affected PRMT1 expression, with an average of 77% remaining PRMT1 mRNA level. These findings suggest that the IpA mechanism characterized in this study is distinct from the previous two studies on EXO59 and PRMT1.

The authors did not perform any skin function assay following CPSF loss and premature differentiation.

To address these concerns, we performed progenitor competition assay to evaluate the regenerative capacity of keratinocytes following CPSF loss. We also performed Nile Red staining to assess the barrier function of regenerated epidermal tissue. Details of these experiments are included on page 2 of this letter.

CPSF polyadenylation was previously shown to regulate cell differentiation (Singh et al., 2018 Nat Commun), and this is well established for GRHL3 in keratinocyte progenitors. The ExoS9 and PRMT1 papers (Refs 2 and 10) are to some extent similar skin differentiation stories, showing a GRHL3-dependent differentiation mechanism. Therefore, this manuscript lacks novelty.

The Singh et al. 2018 paper¹⁰ cataloged lpa sites in the immune cells, yet this paper did not describe mechanisms controlling lpa usage. Our study advanced the field in several ways. *First*, we profiled genome-wide lpa site usage in keratinocytes and revealed diverse lpa site usage between different cell types. *Second*, we uncovered that CPSF downregulation influences both keratinocyte differentiation and lpa usage. *Third*, we demonstrated using CRISPR KO experiments that GRHL3 lpa plays a role in suppressing the full-length GRHL3 mRNA expression, a distinct mechanism from previously described protein truncation caused by lpa. *Fourth*, we identified HNRNPA3 as an interacting protein that cooperates with CPSF to suppress GRHL3 expression and keratinocyte differentiation.

GRHL3 is strongly upregulated in differentiated keratinocytes. How GRHL3 is suppressed in progenitor maintenance remains incompletely understood. The PRMT1 paper¹¹ showed that CSNK1a1 is essential to maintain PRMT1's genomic binding to GRHL3. How PRMT1 genomic binding represses GRHL3 expression is unclear. We have examined CPSF expression in the context of PRMT1 knockdown and found that the subunits were not significantly altered (data included in Supplementary Fig S3a). EXOSC9, on the other hand, binds directly to GRHL3 mRNA to mediate its degradation in progenitors¹². Thus our findings of GRHL3 lpa represent a distinct mechanism of modulating GRHL3 expression, which is different from the actions of PRMT1 and EXOSC9.

• Point by Point Response to Reviewer #2

We appreciate the constructive suggestions from Reviewer #2. The details are listed below. (*Reviewer's original comments are in Gray, responses are in Black, additions of new data are indicated in Blue*).

In this manuscript, Chen and colleagues studied differential intronic polyadenylation (lpa) during human keratinocyte differentiation. They further identified a prominent lpa site within the first intron of GRHL3, an important TF for epidermal differentiation, and demonstrated the downregulation of CPSF or the deletion of the GRHL3 lpa site increased the expression of full-length GRHL3 in progenitor keratinocytes. They further identified a number of proteins interacting with CPSF including hnRNPA3, which plays a role in alternative splicing. They propose a mechanism by which CPSF and hnRNPA3 function to reduce the production of full-length GRHL3 mRNA in progenitor keratinocytes by promoting alternative splicing and intronic polyadenylation within the first intron. Overall, this is an interesting study which provides new insights into how differential lpa sites usage plays a role in epidermal differentiation. That being said, many more analysis and detailed studies should be carried out to strengthen the study in the potential revision.

I have the following comments:

1. In Fig 1, they performed 3'READS+, an improved 3'end sequencing technique, with undifferentiated and differentiated human keratinocytes. Although they mostly focus on lpa sites, they should also identify differentially used polyA sites within the 3'UTRs and determined 1) how many genes show differentially used polyA sites in canonical 3'UTRs; 2) whether the use of proximal sites is a feature for epidermal differentiation.

We agree that it is important and interesting to examine the PolyA usage in 3'UTR. In total, we identified 2727 genes that are associated with multiple PolyA sites (≥ 2) in 3'UTR in keratinocytes. To determine if 3'UTR lengthening or shortening could be a feature during keratinocyte differentiation, we calculated the fraction of distal-site usage within the 3'UTR for these 2727 genes. We identified that 668 out of these 2727 genes are associated with altered usage of the distal site (fold change > 1.5). These include 457 genes with decreased usage and 211 genes with increased usage of their distal sites in differentiation. These data suggest that both lengthening and shortening of 3'UTR occurs during keratinocyte differentiation, although shortening occurs more

frequently than lengthening. This trend is similar to the observation in spermatogenesis¹³. These new data are now included as Supplementary Fig. 1g-i.

2. Among 2703 lpA sites, they identified 610 differentially used sites. Motif analysis should be performed to determine whether these 610 sites have unique features that promote the differential usage (over- or under-used during differentiation), in comparison to the unchanged lpA sites as well as to changed and unchanged canonical 3'UTR sites.

We appreciate this great suggestion. Using MEME motif search, the canonical motif AAUAAA is confirmed as the top enriched RNA motif in all lpA sites (E value: 3.3e-199) and differentially used lpA sites (E value: 1.9e-165). Pair-wise differential motif search between differential vs unchanged lpA sites did not reveal significantly changed motif with E value <0.05. These findings indicate that the differentially used lpA sites are not associated with significantly altered enrichment of the canonical motif. In addition, the lack of significantly changed motif, between differential lpA and all lpA sites, is actually consistent with our findings in the case of HNRNPA3. We found that HNRNPA3 cooperates with CPSF to control *GRHL3* lpA, but not other lpA sites such as those associated with *ALOX15B* and *CRBN*, suggesting that the regulation of these differential lpA sites may require diverse mechanisms.

Comparing the differential lpA sites with differential 3'UTR sites or with unchanged 3'UTR sites, we identified the same top motif related to RBM6 in both comparisons. The E values were 1.4e-010 and 4.2e-18, suggesting the representation of this motif occurs in a small subset of targets. We also found the enrichment of CNOT4 motif (E value: 6.2e-009) from the comparison between lpA versus unchanged 3'UTR. Both RBM6 and CNOT4 are expressed in keratinocytes, based on our RNA-seq data. These findings suggest that RBM6 and CNOT4 might influence the usage of lpA sites in a subset of targets.

3. Similarly, they found 379 sites that are shared between keratinocytes and immune system. They should determine 1) whether the same host gene used different or identical lpA sites; 2) whether the small overlap is due to the differential expression of the host genes or the differential usage of lpA sites even when the same host genes are expressed. If the latter case is true, do the differentially used lpA in keratinocyte vs immune systems have different motifs?

We agree with the reviewer that the low overlap (17% of KC lpA) between keratinocytes and the immune system is intriguing. We first compared these lpA sites with the Polyadenylation database PADB3. 27% of the lpA sites from our study overlap with PADB3; the lpA sites identified from the immune system only have 12% overlap. We also compared the genes associated with these lpA sites between keratinocytes and the immune system, and found that the degree of overlap increased from 17% to 45.3%. For example, 3 lpA sites in *AGO3* were found in keratinocytes, and 4 lpA sites in *AGO3* were found in the immune cells. Only 2 of these overlap. These data suggest that different lpA sites are used even within the same genes in these 2 cell types.

We also performed differential motif analysis between the lpA sites associated with these two cell types. The top motif discovered by DREME is CxCC (E-value: 3.3e-44). It is a short motif associated with *SRSF2*, *LIN28A* or *HNRNPH2*. All these 3 genes are expressed in keratinocytes. The second motif C/GAGxC/G (E-value: 9.7e-30) is not associated with a known regulator. These data suggest that different RNA motif recognition does occur between the lpA usage in these two cell types, although the exact identity of the regulators will need to be validated. These data are now included as supplementary Fig. 1c-f.

4. They found 66.4% of lpA sites are preferentially used in undifferentiated cells and they showed a few examples including ESPN and IQCK. However, is there any functional relevance of these 405 genes during epidermal differentiation? Do ESPN and IQCK play any role in epidermal differentiation? In other words, is there any global impact of differential lpA usage other than the case of GRHL3?

We thank the reviewer for pointing this out. A total of 373 genes are associated with the 428 differentially used lpA sites from this study. A subset of these genes is directly involved in establishing epidermal barrier function. These known differentiation effectors include several members of the lipoxygenase family *ALOXE3*, *ALOX12B*, and *ALOX15B*, which are essential for the barrier function of epidermal tissue. In addition to these classic examples of well-characterized differentiation markers, a number of other genes are predicted to be relevant to keratinocyte differentiation. For example, *ESPN* encodes an actin-bundling protein. The expression of *ESPN* is highly specific for skin and testis, with minimal expression in other tissue types such as fat, heart, brain, and pancreas. *IQCK* interacts with EF-hand proteins, which can bind calcium, a stronger inducer of keratinocyte differentiation. Thus these lpA sites are associated with the multifaceted function of keratinocytes,

although many of them have not yet been well characterized in this system. We have updated the manuscript to better explain the roles of these genes.

5. All Western results in Figs. 1-3, 6 should be repeated and quantified.

We thank the reviewer for bringing up this important aspect to ensure reproducibility. We have replicated these western blots pointed out by the reviewer. All the new western blots added during this revision are also replicated. Quantification and standard deviation are labelled below the western blots.

6. In Fig. 2 studies, they knocked down CPSF1, observed downregulation of multiple CPSF subunits at the protein level and detected increased expression of epidermal expression. To confirm the changes of other CPSF subunits are at the protein level, they should also measure mRNA by qPCR. They nicely showed the difference in epidermal differentiation in an organotypic system when knocking down CPSF. However, more thorough analysis should be performed in this system such as whether KD CPSF leads to the reduction or deletion of basal stem cells, whether basement membrane is affected and whether proliferation/cell death is affected.

We thank the reviewer for these suggestions. we have included qPCR showing that CPSF1 knockdown did not significantly alter the expression of other CPSF subunits at the mRNA level. These new data are now included as Supplementary Fig. 2b.

We have performed several additional experiments to further analyze CPSF's role in tissue regeneration. To test the regenerative capacity of keratinocytes, we performed a progenitor competition assay, which shows that CPSF knockdown significantly impaired keratinocyte proliferation and tissue regenerative capacity. We also included additional analyses comparing epidermal tissues regenerated with just CPSFi or CTRLi keratinocytes. Using Nile Red staining, we found that the lipid layer of CPSFi is thicker, which is consistent with the upregulation of genes related to lipid processing such as ALOX15B with CPSF knockdown. We performed TUNEL assay, and found no significant difference between CPSFi versus CTRLi tissues, suggesting cell death is not the major contributor to the hypoplastic tissue regeneration. These new data are now included as Fig. 2b,c, and Supplementary Fig. 2d-f. [Redacted]

7. They used both RNAi and CRISPRi to repress CPSF1 to reduce off-target effect. A head-to-head comparison of RNA-seq datasets derived from both inhibitory studies should provide the most critical insights into consistency and identify commonly altered genes independent of experimental approaches to inhibit CPSF.

We agree that directly intersecting these two RNA-seq data sets is a better way to present the data. We have reorganized the figures in the manuscript accordingly. The direct comparison is now included in Fig.2. A total of 739 target genes are shared in both approaches. The enriched GO terms of these genes are consistent with the previous findings. These new data are now included as Fig. 2h-j.

8. It is unclear why the H73A CPSF3 mutant can function as dominant negative. Does the mutant bind to nascent RNA but fail to process them? If that's the case, could they observe many mRNAs w/o polyA tail and can these abnormal mRNA be stabilized? It seems this approach will interfere with normal mRNA metabolism rather than inhibiting lpa alone. Unless they can provide more robust characterization such as 3'READS+ results, I am not convinced that these data will provide a clear argument for their role in suppressing premature differentiation.

We agree with the reviewer that the inclusion of this construct raises more questions than clearly supporting the role in alternative polyadenylation. These results, although interesting, do not closely fit within the scope of this story. We have removed this specific panel accordingly.

9. They showed that 399 out of 610 differentially used lpa sites during epidermal differentiation is also affected by the downregulation of CPSF. This suggests that these sites are more sensitive to the levels of CPSF than the remaining 211 sites. They should perform motif analysis to determine if these sites harbor some unique sequences. In addition, they should also analyze whether mRNA levels are higher or lower for these 399 sites when comparing to the unchanged sites.

We appreciate this great question. We were also intrigued by the fact that not all differential lpa sites were affected by CPSF knockdown. Between CPSF-dependent versus CPSF-independent lpa sites, no significantly enriched motif was uncovered using MEME. The mRNA levels of the genes in these two groups are

comparable (new data included as supplementary Fig. 4a), suggesting that expression level is not a direct contributing factor distinguish these two groups. On the other hand, we noticed that several genes encoding other complexes involved in the polyadenylation process are also differentially regulated in keratinocyte differentiation. For example, CSTF2 is strongly downregulated, while PCF11 is upregulated (new data included as supplementary Fig. S1j). Both CSTF2 and PCF11 have been previously implicated in controlling alternative polyadenylation in other systems. Thus it is likely that the CPSF-independent IpA sites could be influenced by other regulators involved in the cleavage and polyadenylation process.

10. In Fig. 5, they studied the GRHL3 site in more details with CRISPR deletion. However, I'm not sure whether Fig. 5b clearly demonstrated the deletion pattern. They should use either deep sequencing or at least sequence 10~20 clones of the DNA fragment rather than using the mixture. More importantly, they should try to derive single clones from these CRISPR experiments if possible. They could have a much better understanding of how the deletion of this site affects GRHL3 expression and keratinocyte in general. In its current form, they only have a mixture of cells with ~50% deletion efficiency, and this doesn't count for heterozygous deletion events. To show the increase of GRHL3, they should perform Western quantification at minimum.

We agree with the reviewer that CRISPR KO in keratinocytes is heterogeneous. Primary human keratinocytes have a limited life span in cell culture conditions. Thus it is technically challenging to isolate and expand individual clones of keratinocytes, unless KO of a specific genomic element could provide growth advantage. In order to circumvent this technical barrier, we analyzed publicly available RNA-seq data and found that this GRHL3 IpA site also used in a few cancer cell lines including HCT116. We screened a total of 60 clones that were expanded from HCT116 cells, which were transduced with Cas9 and two sgRNAs flanking the GRHL3 IpA site. We identified 1 clone with deletions in both alleles (KO) and 9 clones with deletions in one of the two alleles (HET). We further performed Sanger sequencing to closely examine the deletion status of the KO and 2 representative HET clones. The deletion of GRHL3 IpA was confirmed in all 4 alleles we sequenced. In particular, GRHL3 mRNA expression was increased ~20 times in the KO clone (new data included as Supplementary Fig. 5c-h). These data provide another piece of evidence, using a different type of cells, to validate the effectiveness of the KO strategy. (The GRHL3 expression level is lower in HCT116 cells than keratinocytes. Based on our qRT-PCR, the CT value of GRHL3 in HCT116, even in KO, is 5 CT higher than undifferentiated keratinocytes. These findings suggest that GRHL3, although expressed in HCT116 cells, may not play a leading role in controlling the growth of these immortalized cells. We have compared the growth of HCT116 KO and control cells using clonogenicity assay, and did not observe drastic differences between these two conditions.)

We agree with the reviewer that it is important to validate GRHL3 protein using western blotting. A previously validated rabbit antibody⁵ is now discontinued from the vendor. We purchased and tested a total of 4 other antibodies raised against GRHL3, which are currently commercially available. As GRHL3 is strongly upregulated during keratinocyte differentiation, our first step of validation was to compare the pattern between undifferentiated and differentiated keratinocyte lysate. (According to our RNA-seq, GRHL3 is upregulated ~30 fold in differentiation.) Only 2 of these 4 antibodies showed an increased band corresponding to the size of GRHL3 in the differentiation condition. The level of GRHL3 in undifferentiated keratinocyte is below the sensitivity of the detection, with a number of non-specific bands. In the heterogeneous population of keratinocytes with GRHL3 KO (~4-5 fold upregulation, a lot less than differentiation), we could see a very weak band appearing yet the data is not of publication-quality. We have also tried the HCT116 KO line. Consistent with the lower abundance based on our findings from qPCR, the GRHL3 band, although detectable in differentiated keratinocytes, is below the detection sensitivity limit of these antibodies. The quality and sensitivity of commercial antibodies are a limitation for this test. We did find that a number of differentiation marker genes, downstream to GRHL3, were upregulated in keratinocytes with CRISPR disruption of the IpA sites. These include the GRHL3 direct targets SBSN and CRCT1 (new data included as Fig 5e,j,k), supporting the model that GRHL3 protein is upregulated in these conditions to drive differentiation. The list of tested antibodies are included in Table R2 in Appendix II of this letter.

11. In Fig. 7, they aimed to identify CPSF-interacting proteins, whose KD synergize the control of GRHL3 IpA. They identified hnRNPA3 as a candidate. They went on to show that hnRNPA3 may affect an alternative splicing event within the same intron of GRHL3. However, it is not clear to me why the alternative splicing could further enhance the IpA usage by GRHL3. Does the recruitment of hnRNPA3 to the intron directly increase the binding of CPSF to the IpA or the splicing itself recruits CPSF? Despite the provided evidence, it remains how the synergy between hnRNPA3 and CPSF is generated. Regardless the potential mechanism, I am also curious about

whether the hnRNPA3/CPSF duo only functions on the GRHL3 site or other sites may also be affected.

We apologize that the interaction between HNRNPA3 and CPSF was not well explained in the previous version. We have revised the manuscript with the following updates. *First*, we generated RNA-seq data of HNRNPA3 knockdown. To our surprise, more genes (1490 total) were significantly affected in HNRNPA3 knockdown than CPSF knockdown. 69% of these HNRNPA3 targets overlap with calcium-induced differentiation signature, while only 20% of these HNRNPA3 targets overlap with CPSF (new data included as Fig. 7b-d). These data suggest that HNRNPA3 has CPSF-dependent and CPSF-independent roles in suppressing differentiation in epidermal progenitors. *Second*, we performed co-immunoprecipitation between CPSF and HNRNPA3 with RNase treatment, and we found that the interaction was not sensitive to RNase. We also confirmed that CPSF co-immunoprecipitated HNRNPA3 in undifferentiation but not in differentiation condition (new data included as Fig. 6e,f). These data suggest that direct protein-protein interaction occurs between CPSF and HNRNPA3 in the progenitor state but not in differentiated keratinocytes. *Third*, we have updated the figures to show that HNRNPA3 does not cooperate with CPSF to influence other Ipa sites such as the ones associated with ESPN and CRBN (new data included as Supplementary Fig. 6d,e). *Fourth*, we have performed motif searches in the regions upstream to the GRHL3 Ipa site. Although HNRNPA3 is currently under-characterized, and its motif is not in the collection of RNA motif databases, the motif of HNRNPA1 is well-characterized. HNRNPA1 and HNRNPA3 share 94% similarity in their RNA-binding domain. Within 2kb upstream of the GRHL3 Ipa, we identified 3 HNRNPA1 motif sites. These findings suggest that HNRNPA3 can bind to the nascent RNA upstream of the Ipa site. *Fifth*, we have simplified the splicing assay to better clarify the two possibilities of alternative splicing (Fig. 7j-m). Last but not least, we have included a better explanation of why alternative splicing can enhance Ipa usage in the discussion section. In brief, it has been demonstrated that multiple copies of HNRNPA1 can bind and spread on nascent RNA with two functions: altering its secondary structure and displacing splicing-promoting proteins such as the serine/arginine (SR)-rich-family proteins. We speculate that the cooperation between HNRNPA3 and CPSF could have two roles. HNRNPA3's binding to the intron can suppress the binding of splicing activators and inhibits the splicing between exon1 and exon2; HNRNPA3's physical interaction with CPSF may further stabilize CPSF binding to its canonical motif to promote cleavage and polyadenylation.

• Point by Point Response to Reviewer #3

We appreciate the thoughtful and constructive feedback from the reviewer. In particular, we are thankful for the suggestion to look for the mechanisms controlling CPSF downregulation during differentiation, and to improve the figures related to CPSF's interaction with HNRNPA3. Please find the details listed below (*Reviewer's original comments are in Gray*, responses are in Black, additions of new data are indicated in Blue).

In the manuscript by Chen et al, the authors explore the mechanism of keratinocyte differentiation and the role that the CPSF complex plays in regulating altered cleavage and polyadenylation. The authors first utilize a well-characterized primary keratinocyte human cell line and perform both RNA-seq and 3READS+ both before (UD) and after (DF) differentiation. They observe significant reduction in the usage of intronic poly(A) sites (Ipa) sites as cells differentiate. They then determine that members of the core CPSF complex are being downregulated as cells differentiate. Using two distinct approaches (RNAi and CRISPRi), they then determine that downregulation of CPSF1 alone is sufficient to stimulate differentiation and drive down usage of Ipa sites. They then use CRISPR to drill down on the importance of a specific Ipa site within GRHL3 and show that deletion/mutation of this site increases its expression. Finally, they conduct IP MS/MS analysis of CPSF complexes before and after differentiation to identify interacting proteins that are specific to the UD state and focus on one of those factors, hnRNPA3, to determine its functional relevance. Overall, this is a nice study that has many novel findings. In fact, I have just a series of minor concerns for Figures 1-5.

Having said this, I have issues with the data presented in Figures 6 and 7. It almost seems that these two figures could be removed and the overall strength of the study goes up. My first issue is the simple logic of conducting the mass spec. The authors state that 'differential levels of CPSF proteins...raises the possibility of differential protein interactions in the two conditions...why, this doesn't seem to be logical or at least the logic needs to be presented better. Even more confusing, the authors show that many of these 'CIPs' are also downregulated just like the CPSF proteins. So, ultimately, the levels of CPSFs and CIPs go down together making it strange to me that a differential association would occur.

We apologize that the rationale for conducting the mass spec experiment was overly simplified in the previous version. We also agree that the CPSF-interacting proteins, identified from IP-mass spec, could be mediated by RNA. To address these concerns, we made several changes to the manuscript. We revised the results section to better explain the goal of this targeted screen, which was to identify regulators that can enhance CPSF-mediated control of GRHL3 IpA. We de-emphasized the mass spec data, moved them to supplementary data and specified in the manuscript that these were a “pilot mass spec” experiment. We also included CSTF2, the most downregulated gene among all the genes encoding other key complexes involved in cleavage and polyadenylation (PAP, CSTF, CFII, CFI), in our targeted screen. Fig. 6 and Supplementary Fig 6 are substantially revised to incorporate these changes.

We agree that the previous quantification of RNA binding proteins, based on RNA-seq data, was not ideal. To address this, we performed western blotting to quantify HNRNPA3 protein expression in keratinocyte differentiation. The protein is expressed in both undifferentiated and differentiated keratinocytes (new data included as Supplementary Fig.6 c). We speculate that specific post-translational modifications might also be involved to mediate the interactions between CPSF and HNRNPA3 in undifferentiated keratinocytes.

To begin with, the methods suggest that the mass spec was not done using RNAase treatment and it should. One of the supplemental data uses RNAse in the validation but the initial mass spec should be. The authors confirm mass spec association using specific IPs but they appear to be only in the UD cell lysates. Shouldn't the authors validate the findings of differential association so IPs from both UD and DF be done in comparison?

We thank the reviewer for pointing out. Fig. 6 is updated to include CPSF-HNRNPA3 co-IPs in both undifferentiation and differentiation conditions as well as with RNase treatment. We confirmed that HNRNPA3 was not co-immunoprecipitated by CPSF1 in the differentiation condition, and the interaction in the undifferentiation condition was retained with RNase treatment. These data suggest that the interaction between CPSF and HNRNPA3 does not require the bridge function from RNA.

They focus on hnRNPA3 but the experiments raise more questions than they answer. The authors need to do both RNA-seq to address splicing changes and 3'READS+ once hnRNPA3 is knocked down.

To address the concerns on splicing changes, we have examined several aspects to better clarify the roles of HNRNPA3 in splicing. We have improved the qPCR strategy to quantify the two alternative splicing events, either “skipping” or “inclusion” of the hidden exon. HNRNPA3 knockdown increased the ratio of “skipping” vs “inclusion”, suggesting that HNRNPA3 suppresses the junction between exon1 and exon2 (new data and illustration included as Fig. 7i,j). [Redacted]

HNRNPA1 and HNRNPA3 share 94% similarity in their RNA-binding domain, suggesting that these two proteins might use the same motif to bind to RNA. The RNA motif for HNRNPA1 has been identified in the literature and is included in the RBPmap database¹⁴. Within 2kb upstream of the GRHL3 IpA site, we identified 3 motifs corresponding to HNRNPA1, suggesting that HNRNPA3 could bind to the nascent RNA upstream of the IpA site. we speculate that the cooperation between HNRNPA3 and CPSF may impact splicing and IpA usage in two ways. HNRNPA3's binding to the intron can suppress the binding of splicing activators, similar to what had been demonstrated for HNRNPA1¹⁵, to inhibit the splicing between exon1 and exon2. HNRNPA3's physical interaction with CPSF may further stabilize CPSF binding to its canonical motif to promote cleavage and polyadenylation. We have revised the discussion section to include our current understanding of how alternative splicing could enhance IpA usage.

The RNA-seq data of HNRNPA3 knockdown also revealed 1490 HNRNPA3 targets. 69% of these HNRNPA3 targets overlap with calcium-induced differentiation signature, while only 20% of these HNRNPA3 targets overlap with CPSF. These data suggest that HNRNPA3 has CPSF-dependent and CPSF-independent roles in suppressing differentiation in epidermal progenitors. These new data are included as Fig. 7b-d.

To clarify HNRNPA3's role in controlling IpA, we quantified the usage of IpA sites in CRBN and ALOX15B comparing HNRNPA3 knockdown, CPSF knockdown, as well as their double knockdown. The results were quite different from the case of GRHL3 IpA. In the case of CRBN, HNRNPA3 knockdown decreased IpA usage, however the IpA usage was not drastically further decreased in the double knockdown condition. In the case of ALOX15B, HNRNPA3 knockdown alone further increased IpA usage. Thus, HNRNPA3 does not cooperate with CPSF in controlling all the CPSF dependent IpA sites. This is consistent with the findings from RNA-seq that HNRNPA3 has CPSF-independent roles in controlling gene expression in keratinocytes.

Finally, it is this reviewers opinion that the authors left the most interesting question on the table: How does the CPSF complex get downregulated?? Unfortunately, there is no data presented on this question.

The reviewer raised an excellent question regarding the mechanism underlying CPSF downregulation in keratinocyte differentiation. We investigated several different aspects to address this. We first asked if impaired proliferation and induced differentiation is sufficient to downregulate CPSF. Knocking down PRMT1, an essential regulator for sustaining proliferation and suppressing differentiation in epidermal progenitors, did not affect CPSF expression. Prolonged culture of primary keratinocytes also induced differentiation and impaired proliferation marker expression; yet CPSF expression was only minimally affected. These data suggest that the downregulation of CPSF in keratinocyte differentiation is not simply due to impaired proliferation and induced differentiation. To identify specific transcription regulators that may function upstream to control CPSF expression, we searched publicly available ChIP-seq data sets for factors that bind to the promoter region of CPSF1. We identified MYC as a candidate. MYC is downregulated during keratinocyte differentiation. MYC knockdown significantly downregulated CPSF expression in progenitor keratinocytes. These data revealed MYC as a key regulator required for sustaining CPSF expression in undifferentiated keratinocytes. These new data are included as Fig. 3 and Supplementary Fig. 3.

Other issues:

1) The 232989 polyA sites were distributed over how many genes?

To increase the robustness of our data analysis, we used a more stringent cutoffs and an updated annotation system. Each PAS had to have a total of ten total counts from both libraries and an FPU of at least 10% from one library. The previous cutoffs were five counts and 5% FPU. Instead of using Homer annotation which assigns sites to the gene with the nearest TSS, we used GENCODE v17 annotation tables from UCSC table browser. We further removed sites that aligned to more than one gene and were ambiguous. In addition, a 2 fold-change cut off instead of 1.5 fold-change cutoff was used to identify differentially used lpa sites from 3'READS+ data. We now have a total 14625 of PolyA sites that are distributed among 7991 genes in keratinocytes. We have updated both the results and methods section to reflect these changes.

2) The fact that ~10% of all poly(A) sites are intronic poly(A) sites coupled with the fact that very few overlap with other recently published data is potentially concerning. Can you authors provide evidence that internal priming isn't happening or priming to degraded RNA that has been adenylated by the TRAMP complex? Can you authors comment on what percentage of their IPAs have already been annotated? Can the authors also do a MEME analysis to provide evidence as to what percentage of the IPA sites contain polyA motifs upstream? This also would provide additional confidence that the sheer number of IPA sites is a real thing.

We agree with the reviewer that the low overlap (17% of KC lpa) between keratinocytes and the immune system is intriguing. We compared these lpa sites with the Polyadenylation database PADB3, and found 12% of the lpa sites identified immune-system study overlap with PADB3. The lpa sites identified from keratinocytes actually have 27% overlap with PADB3, suggesting that more sites from our study are previously annotated. is actually higher, with 27% of the lpa sites identified from keratinocytes overlap with PADB3. One difference between our approach vs the immune study is the usage of different techniques. While the immune system uses 3'-seq, our study uses 3'READS+ which specifically leverages an LNA to minimize internal priming. MEME motif search further revealed that the canonical AAUAAA motif is highly enriched in all lpa sites (E value: 3.3e-199) as well as in differentially used lpa (E value:1.9e-165). These findings are now incorporated as Supplementary Fig. 1c-f.

3) In figure 1D, it seems to be a peculiar choice to measure CPSF members in comparison to Rpb1. The authors should also consider measuring members of the CstF, CFIm, and CFII complex as these are also considered to be the major players in cleavage and polyadenylation.

We appreciate the reviewer for pointing this out. Quantification of genes encoding these complexes, during keratinocyte differentiation, is now included as Fig. S1j. We have further tested CSTF2 (the most downregulated among these 3 complexes in keratinocyte differentiation) knockdown to determine if it can enhance CPSF's control of GRHL3 lpa. The effect was minimal, and HNRNPA3 had a much stronger effect. These new data are now incorporated as Supplementary Fig. 1j, and Fig. 6b,c.

4) Figure 1j is not overly convincing. The authors should consider additional images to convey their main point.

We apologize that the previous staining was suboptimal. We have updated with an improved image to convey the main point. Raw images are included in the Raw Data file.

5) Figure 6F, the authors should label or at least mention in the legend what percentage of input and pulldown are loaded on the gels.

We appreciate this suggestion, and we have included quantification accordingly in Fig. 6e,f.

Minor

1) Typo 'catalytic-inactivate' should be '-inactive'

We appreciate the reviewer for pointing this out. This specific piece of data was removed during the revision process, considering the input from Review #2. We will be extra careful during the proofreading stage.

References cited in this letter:

1. Klein, R. H. *et al.* GRHL3 binding and enhancers rearrange as epidermal keratinocytes transition between functional states. *PLOS Genet.* **13**, e1006745 (2017).
2. Lopez-Pajares, V. *et al.* A LncRNA-MAF:MAFB Transcription Factor Network Regulates Epidermal Differentiation. *Dev. Cell* **32**, 693–706 (2015).
3. Lee, S.-H. *et al.* Widespread intronic polyadenylation inactivates tumour suppressor genes in leukaemia. *Nature* **561**, 127–131 (2018).
4. Ji, Z., Lee, J. Y., Pan, Z., Jiang, B. & Tian, B. Progressive lengthening of 3' untranslated regions of mRNAs by alternative polyadenylation during mouse embryonic development. *Proc. Natl. Acad. Sci. U. S. A.* **106**, 7028–33 (2009).
5. Zhao, P. *et al.* Grhl3 induces human epithelial tumor cell migration and invasion via downregulation of E-cadherin. (2016). doi:10.1093/abbs/gmw001
6. Sen, G. L., Reuter, J. A., Webster, D. E., Zhu, L. & Khavari, P. A. DNMT1 maintains progenitor function in self-renewing somatic tissue. *Nature* **463**, 563–7 (2010).
7. Rubin, A. J. *et al.* Lineage-specific dynamic and pre-established enhancer–promoter contacts cooperate in terminal differentiation. *Nat. Genet.* **49**, 1522–1528 (2017).
8. Hopkin, A. S. *et al.* GRHL3/GET1 and Trithorax Group Members Collaborate to Activate the Epidermal Progenitor Differentiation Program. *PLoS Genet.* **8**, e1002829 (2012).
9. Kerkhoff, C. *et al.* Novel insights into the role of S100A8/A9 in skin biology. *Exp. Dermatol.* **21**, 822–826 (2012).
10. Singh, I. *et al.* Widespread intronic polyadenylation diversifies immune cell transcriptomes. *Nat. Commun.* **9**, 1716 (2018).
11. Bao, X. *et al.* CSNK1a1 Regulates PRMT1 to Maintain the Progenitor State in Self-Renewing Somatic Tissue. *Dev. Cell* **43**, (2017).
12. Mistry, D. S., Chen, Y. & Sen, G. L. Progenitor function in self-renewing human epidermis is maintained by the exosome. *Cell Stem Cell* **11**, 127–35 (2012).
13. Li, W. *et al.* Alternative cleavage and polyadenylation in spermatogenesis connects chromatin regulation with post-transcriptional control. *BMC Biol.* **14**, (2016).
14. Paz, I., Kosti, I., Ares, M., Cline, M. & Mandel-Gutfreund, Y. RBPmap: a web server for mapping binding sites of RNA-binding proteins. *Nucleic Acids Res.* (2014). doi:10.1093/nar/gku406
15. Okunola, H. L. & Krainer, A. R. Cooperative-Binding and Splicing-Repressive Properties of hnRNP A1. *Mol. Cell. Biol.* **29**, 5620–5631 (2009).

[Redacted]

Reviewers' Comments:

Reviewer #1:

Remarks to the Author:

I appreciate the efforts made by the authors to address the reviewers' comments. Before congratulating them on this successful study, few comments can still be considered to clarify the mechanism(s) described in the paper.

1- Fig. 3: With regards to the role of c-Myc in keratinocyte differentiation, can the authors look for the relationship between c-MYC and GRHL3 and make this figure more comprehensive? Klein et al., PLOS Genetics, 2017 suggested a direct relationship by ChIP-seq and other studies by Georgy et al., J Natl Cancer Inst., 2015 found that GRHL3 is upstream of c-MYC.

2- The GRHL3 antibody from Aviva Systems Biology seems to work for western blotting on cells. Multiple papers have been published using this antibody. The authors may want to validate their experiments using it.

3- The GRHL3 target genes and genes involved in differentiation (e.g. S100A8/A9) or terminal differentiation should be distinguished clearly.

Reviewer #1: Charbel Darido

Reviewer #2:

Remarks to the Author:

In the revised manuscript, Chen et al., have significantly improved their studies by performing additional analysis, quantification, and more robust CRISPR deletion of the IpA site within GRHL3 intron1. Overall, this study has elegantly demonstrated a new role of CPSF1 downregulation and its functional effect on the differential use of an IpA site within GRHL3 intron 1 during human keratinocyte differentiation. Although the differential use of IpA sites has been described in other systems such as cancer cells and muscle regeneration, the mechanism of the differential IpA usage remains unclear. This study should be of interests for the broad audience of Nature Communications. I have the following minor points for additional clarification by the authors.

1. Page 6, line 144, the motif should be AAUAAA.
2. In Fig. 2b-c, the authors performed a competitive regeneration assay. Based on the image, it appears that CPSF1 knockdown cells (H2b-mCherry) are not only reduced globally but also specifically depleted from the basal layer. I suggest the authors perform a more careful quantification on these experiments.
3. Page 15, line 374, the authors stated "HNRNPA3 target genes and CPSF1 core target genes". Because the authors didn't provide evidence for direct binding and regulation of these genes by HNRNPA3 or CPSF1, it should be more appropriately referred as genes affected by the downregulation of HNRNPA3 and CPSF1.
4. The authors provided some evidence that co-knockdown of HNRNPA3 and CPSF1 has a synergistic effect on the expression of GRHL3. Furthermore, the deletion of ~1kb intronic region by sgRNA2/3 results in a strong increase of GRHL3 in HCT116 cells. The authors should comment on whether the deletion of ~1kb region removes both the IpA and the HNRNPA3 binding site (putative), and whether the drastic increases in GRHL3 expression in the deletion reflect both mechanisms.
5. HNRNPA3 protein expression is slightly reduced in differentiated keratinocytes (Fig. S6c). The authors should clearly state it in the text so the readers may appreciate the reduced interaction between CPSF1 and HNRNPA3 in the differentiated cells is likely the result of the reduced expression of both proteins.

Reviewer #3:

Remarks to the Author:

I commend the authors with their efforts. I believe that my previous critiques were somewhat demanding but I am extremely pleased with how the manuscript has evolved. At this point, I am supportive of publication.

Point-by-Point Response to Reviewers' Comments

Reviewer #1 (reviewer's comments are included in gray):

1- Fig. 3: With regards to the role of c-Myc in keratinocyte differentiation, can the authors look for the relationship between c-MYC and GRHL3 and make this figure more comprehensive? Klein et al., PLOS Genetics, 2017 suggested a direct relationship by ChIP-seq and other studies by Georgy et al., J Natl Cancer Inst., 2015 found that GRHL3 is upstream of c-MYC.

We appreciate these suggestions!

The paper published by Georgy et al (J Natl Cancer Inst., 2015) is very interesting! According to this paper, GRHL3 Knockout in tongue epithelium increased Myc expression at the protein level, but not at the mRNA level. Several recent papers demonstrated that MYC protein stability is controlled by phosphorylation at Ser62 (ERK, stabilize MYC protein) and Thr58 (BRD4, destabilize MYC protein). Thus, GRHL3 loss may indirectly increase ERK activity and/or decrease BRD4 activity to promote MYC stability in the KO tongue epithelium tissue. Although the regulation of MYC protein stability is outside the scope of this paper, these pieces of information are very helpful for us to better understand the complexity of epithelial tissue differentiation.

The GRHL3 ChIP-seq data published in Klein et al 2017 were generated using undifferentiated keratinocytes, where GRHL3 expression is very low as compared to differentiated keratinocytes. We did download and process the raw ChIP-seq data published in this paper. There does not appear to be strong enrichment of GRHL3 signal that is clearly above the background near the promoter of c-MYC in undifferentiated keratinocytes. GRHL3 ChIP-seq data in the differentiated keratinocytes, although published, are not available as raw data in GEO. Since GRHL3 knockout tissue did not alter MYC expression at the mRNA level, as demonstrated by Georgy et al (J Natl Cancer Inst., 2015), GRHL3 is not very likely to directly control MYC mRNA expression at the transcription level through chromatin binding.

2- The GRHL3 antibody from Aviva Systems Biology seems to work for western blotting on cells. Multiple papers have been published using this antibody. The authors may want to validate their experiments using it.

Thank you for letting us know! We ordered the antibody (Cat# OACA10235, LOT # F1013A) from Aviva Systems Biology. We were very hopeful, as a really nice western blotting image is included on the company website, showing a clean band with the correct molecular weight of GRHL3 (68KDa) using lysates from 3 different cell types. This antibody “may replace item sc-101968 from Santa Cruz Biotechnology”, as suggested by the product sheet. However, this specific tube of antibody we received, after waiting 11 days for the company to ship this antibody to us, did not work as well as we hoped. We first tested this antibody using undifferentiated and differentiated keratinocyte lysate in urea lysis buffer, and found that this antibody strongly cross-reacted with several non-specific bands around 50 KDa. In our attempts to optimize this antibody, we tried a different set of keratinocyte lysate in RIPA buffer as well as cutting away the lower half of the blot containing the non-specific bands. The noise level was still very high. We further tested keratinocyte lysate of control versus CPSF knockdown and KO keratinocytes. We were not able to detect strong/convincing bands at the correct size. So far, we have purchased and tested 5 different kinds of GRHL3 antibodies from different vendors, as a highly cited GRHL3 antibody was discontinued. The quality of these antibodies we tested were suboptimal.

3- The GRHL3 target genes and genes involved in differentiation (e.g. S100A8/A9) or terminal differentiation should be distinguished clearly.

Thank you for the suggestion! During keratinocyte differentiation, S100A8 and S100A9 are considered as mid-differentiation markers; SBSN and CRCT1 are considered as late-differentiation markers. We have specified this in our manuscript.

Reviewer #2 (reviewer's comments are included in gray):

In the revised manuscript, Chen et al., have significantly improved their studies by performing additional analysis, quantification, and more robust CRISPR deletion of the IpA site within GRHL3 intron1. Overall, this study has elegantly demonstrated a new role of CPSF1 downregulation and its functional effect on the differential use of an IpA site within GRHL3 intron 1 during human keratinocyte differentiation. Although the differential use of IpA sites has been described in other systems such as cancer cells and muscle regeneration, the mechanism of the differential IpA usage remains unclear. This study should be of interests for the broad audience of Nature Communications. I have the following minor points for additional clarification by the authors.

1. Page 6, line 144, the motif should be AAUAAA.

Thank you so much for catching this! Apologies for this typo. It is now updated.

2. In Fig. 2b-c, the authors performed a competitive regeneration assay. Based on the image, it appears that CPSF1 knockdown cells (H2b-mCherry) are not only reduced globally but also specifically depleted from the basal layer. I suggest the authors perform a more careful quantification on these experiments.

That is a great suggestion! We have further quantified the upper (more differentiated) versus lower (including the basal progenitor layer) half of the epidermal tissue using ImageJ. Indeed, the bottom half showed a more striking difference ($p=1.76 \times 10^{-20}$, t-test, two tailed) than the upper half ($p=5.09 \times 10^{-5}$, t-test, two tailed). These new data are now incorporated as supplementary figures 2c,d.

3. Page 15, line 374, the authors stated "HNRNPA3 target genes and CPSF1 core target genes". Because the authors didn't provide evidence for direct binding and regulation of these genes by HNRNPA3 or CPSF1, it should be more appropriately referred as genes affected by the downregulation of HNRNPA3 and CPSF1.

We agree with the reviewer that it is more appropriated to rephrase this sentence. This statement is now updated as suggested by the reviewer.

4. The authors provided some evidence that co-knockdown of HNRNPA3 and CPSF1 has a synergistic effect on the expression of GRHL3. Furthermore, the deletion of ~1kb intronic region by sgRNA2/3 results in a strong increase of GRHL3 in HCT116 cells. The authors should comment on whether the deletion of ~1kb region removes both the IpA and the HNRNPA3 binding site (putative), and whether the drastic increases in GRHL3 expression in the deletion reflect both mechanisms.

That is a very good point! Based on the Sanger sequencing data, this CRISPR KO strategy only deleted the AAUAAA site, without removing any of three putative HNRNPA3 binding sites upstream to the AAUAAA site. We have clarified this in the "Discussion" section of the manuscript.

5. HNRNPA3 protein expression is slightly reduced in differentiated keratinocytes (Fig. S6c). The authors should clearly state it in the text so the readers may appreciate the reduced interaction between CPSF1 and HNRNPA3 in the differentiated cells is likely the result of the reduced expression of both proteins.

Thank you for the suggestion! We have added this statement in the results section, immediately after mentioning Fig. S6c.

Reviewer #3 (reviewer's comments are included in gray):

I commend the authors with their efforts. I believe that my previous critiques were somewhat demanding but I am extremely pleased with how the manuscript has evolved. At this point, I am supportive of publication.

We really appreciate the suggestions from this reviewer, who helped to make this manuscript a better contribution to the field!